# Differential regulation of the eicosanoid biosynthesis pathway in response to *Enterocytozoon hepatopenaei* infection in *Litopenaeus vannamei*

Wananit Wimuttisuk[1]*, Pisut Yotbuntueng[1], Pacharawan Deenarn[1], Punsa Tobwor[1], Kamonluk Kittiwongpukdee[2,3], Surasak Jiemsup[1], Rapeepun Vanichviriyakit[2,3], Chanadda Kasamechotchung[4], Suganya Yongkiettrakul[1], Natthinee Munkongwongsiri[1,2¤], Siriwan Khidprasert[4], Vanicha Vichai[1]

1 National Center for Genetic Engineering and Biotechnology (BIOTEC), National Science and Technology Development Agency (NSTDA), Khlong Luang, Pathum Thani, Thailand, 2 Center of Excellence for Shrimp Molecular Biology and Biotechnology (Centex Shrimp), Faculty of Science, Mahidol University, Bangkok, Thailand, 3 Department of Anatomy, Faculty of Science, Mahidol University, Bangkok, Thailand, 4 Division of Fisheries, Faculty of Agriculture and Natural Resources, Rajamangala University of Technology Tawan-ok, Bang-Phra, Sriracha, Chonburi, Thailand

¤ Current address: SyAqua Siam Co., Ltd. Sichon district, Nakhon Si Thammarat, Thailand
* wananit.wim@biotec.or.th

## Abstract

The microsporidian *Enterocytozoon hepatopenaei* (EHP) is a highly contagious pathogen that causes severe growth retardation in penaeid shrimp. EHP infection damages the hepatopancreatic tubules, causes hematopoietic infiltration, and recruits granulocytes and inflammatory cells to the shrimp stomach and intestine. In this study, we investigated whether EHP infection induced the eicosanoid biosynthesis pathway in the gastrointestinal tract of the Pacific white shrimp *Litopenaeus vannamei*. Shrimp hepatopancreases, stomachs, and intestines were collected on days 0, 7, and 21 of the EHP cohabitation experiment for analysis. On day 7, the levels of cyclooxygenase (COX) and prostaglandin F synthase (PGFS) enzymes, which catalyze the production of prostaglandins, were elevated in the hepatopancreas of EHP-infected shrimp. The stomach of EHP-infected shrimp also contained higher levels of 12-hydroxyeicosatetraenoic acid (12-HETE) and 12-hydroxyeicosapentaenoic acid (12-HEPE) than the control shrimp. Nevertheless, the most significant impact of EHP infection on day 7 was observed in shrimp intestines, in which the levels of prostaglandin $F_{2\alpha}$ ($PGF_{2\alpha}$), 8-HETE, and four isomers of HEPEs were higher in the EHP-infected shrimp than in the control shrimp. As the EHP infection progressed to day 21, the upregulation of COX and PGFS persisted in the EHP-infected hepatopancreas, leading to increasing levels of $PGF_{2\alpha}$ and 15-deoxy-$\Delta^{12,14}$-prostaglandin $J_2$ (15d-$PGJ_2$). The upregulation of prostaglandins was in contrast with the decreasing levels of HETEs and HEPEs in the hepatopancreas of EHP-infected shrimp.

**Data availability statement:** All relevant data are within the manuscript and its Supporting Information files.

**Funding:** This research has received funding support from the NSRF via the Program Management Unit for Human Resources & Institutional Development, Research and Innovation (https://www.pmu-hr.or.th/en/home) [grant number B05F640184 to W.W.] and the National Science, Research and Innovation Fund, Thailand Science Research and Innovation (TSRI) (https://www.tsri.or.th) [Grant Number FFB670076/0337 to W.W.]. The funders do not play any role in the study design, data collection and analysis, decision to publish, or preparation of the manuscript.

**Competing interests:** The authors have declared that no competing interests exist.

Meanwhile, the stomach of EHP-infected shrimp contained higher levels of prostaglandin $D_2$, $PGF_{2\alpha}$, $15d\text{-}PGJ_2$, and most of the hydroxy fatty acids than the control shrimp. The levels of eicosanoid precursors, namely arachidonic acid and eicosapentaenoic acid, were upregulated in the shrimp gastrointestinal tract collected on days 7 and 21, suggesting that substrate availability contributes to the increasing levels of eicosanoids after EHP infection. Our study provides the first comprehensive analysis of the eicosanoid biosynthesis pathway in response to EHP infection. Moreover, the results indicate that eicosanoids are part of the host-pathogen interactions in crustaceans.

## Introduction

*Enterocytozoon hepatopenaei* (EHP), which rapidly spread across Southeast Asia, is one of the main reasons for the decline in the shrimp aquaculture industry [1–3]. The EHP spore enters the shrimp stomach and passes through the gastric sieve to localize in the hepatopancreas. EHP then replicates in the cytoplasm of hepatopancreatic epithelial cells, causing damage in the hepatopancreatic tubules [2,4]. EHP infection impedes the shrimp's ability to utilize nutrients, resulting in growth retardation that becomes apparent at 2–3 months after the infection [5]. Although the replication of EHP spores occurs in the shrimp hepatopancreas [6], these spores can be detected in the stomach and intestine [7–9]. EHP infection results in the inflammatory responses via hematopoietic infiltration [9]. Additionally, the presence of granulocytes was detected in the hepatopancreas and stomach of EHP-infected shrimp [9]. As eicosanoids are part of the inflammatory pathway, changes in the eicosanoid biosynthesis pathway due to EHP infection was examined in the shrimp gastrointestinal tract.

Eicosanoids are a series of oxygenated derivatives of polyunsaturated fatty acids (PUFAs). These signaling molecules can act both transiently and locally to regulate essential physiological processes, such as immune response, inflammation, and reproduction in most organisms [10–12]. The eicosanoid biosynthesis pathway, which is highly conserved in most vertebrate and invertebrate phyla, consists of two major branches [13]. The prostaglandin biosynthesis pathway produces prostaglandins, prostacyclins, and thromboxanes [14,15]. On the other hand, the lipoxygenase pathway generates hydroxy fatty acids, leukotrienes, lipoxins, and hepoxilins [16,17]. Eicosanoid biosynthesis begins with the cytosolic phospholipase A2 (cPLA2) enzyme, which extracts arachidonic acid (ARA) and eicosapentaenoic acid (EPA) from the phospholipid bilayer [18]. The cyclooxygenase (COX) enzyme catalyzes the conversion of ARA to prostaglandin $H_2$ ($PGH_2$), which is rapidly converted to downstream prostaglandins via prostaglandin synthase enzymes [19]. Once synthesized, these prostaglandins bind to specific cell surface receptors and initiate signaling cascades that regulate various cellular processes [20]. For the lipoxygenase pathway, PUFAs can be oxygenated to produce hydroxy fatty acids by multiple enzymes, including COX, lipoxygenase, and cytochrome P450 [21]. Hydroxyeicosatetraenoic

acids (HETEs) and hydroxyeicosapentaeonic acids (HEPEs) are oxygenated derivatives of ARA and eicosapentaenoic acid (EPA), respectively.

In this study, a cohabitation experiment was performed to induce EHP infection. Shrimp hepatopancreases, stomachs, and intestines were collected on days 7 and 21 of the cohabitation experiment. The analysis revealed that EHP infection resulted in the differential upregulation of the eicosanoid biosynthesis pathway based on the type of organs and infection stages. Moreover, this study is the first to demonstrate that the eicosanoid biosynthesis pathway is induced in response to microsporidian infection in crustaceans.

## Materials and methods

### Ethical statement

The EHP cohabitation assay and sample collection were approved by the Institutional Animal Care and Use Committee at the National Center for Genetic Engineering and Biotechnology, Thailand (BT-Animal 02/2565). All experiments followed the Animal Research: Reporting of In Vivo Experiments (ARRIVE) guidelines and adhered to national and international legal and ethical requirements [22,23].

### EHP cohabitation experiment

Shrimp post-larvae ($n = 1,500$) were purchased from commercial farms and tested for EHP infection following the methodology from Jaroenlak et al. (2016) [24]. Shrimp larvae that tested negative for EHP were raised in a biosecure facility for two months until they reached an average body weight of 3 g per shrimp. Shrimp were then transferred to the shrimp challenge facility and allowed to acclimate for 24 hours.

The control and EHP cohabitation challenge was conducted in separate areas in the shrimp challenge facility. For each treatment, five 600-liter plastic rearing tanks (1.09 m in width x 1.615 m in length x 0.53 m in height; Cosmos Corporation, Thailand) were placed inside a large cement tank. These cement tanks (2.5 m in width x 2.5 m in length x 0.6 m in height) were part of the construction of the shrimp challenge facility. The control and EHP cohabitation challenges were separated by an empty cement tank of the exact dimensions to provide a physical barrier and prevent cross-contamination. One hundred shrimp were placed inside each plastic rearing tank. A nylon net was placed over each rearing tank to prevent the shrimp from escaping. Shrimp were acclimated for 24 hours in the shrimp challenge facility. Eight shrimp from each rearing tank (80 shrimp in total) were collected on day 0 before the cohabitation experiment began.

The cohabitation experiment was performed by placing a plastic basket enclosing healthy shrimp ($n = 10$) or EHP-infected shrimp ($n = 10$) in each rearing tank to establish the control and EHP-infected groups, respectively. The EHP-infected shrimp used in the cohabitation experiment was obtained from a commercial farm where EHP infection has been confirmed using the spore wall protein PCR (SWP-PCR) [3]. The control shrimp with a similar body weight to the EHP-infected shrimp were also obtained from a different commercial farm. The control shrimp were confirmed to be free from EHP using the SWP-PCR. Additionally, these control shrimp were also tested for specific pathogens using the EZEEGENE® nested PCR test kit (BIOTEC, Pathum Thani, Thailand). They must also be free from WSSV, yellow head virus, Taura syndrome virus, and infectious hypodermal and hematopoietic necrosis virus to be used in the cohabitation experiment.

The standard operating practice for the EHP cohabitation challenge was as follows. The staff wore protective footwear that had been decontaminated with 50 ppm iodine before entering the shrimp challenge facility. Shrimp feeding, water quality monitoring, and shrimp sample collection were first performed on the control sample and then on the EHP cohabitation sample. Any staff who entered the EHP-cohabitation zone would not be allowed back in the control region on that given day. Shrimp were fed with commercial feed containing 38% protein twice daily at 9:00 AM and 4:00 PM at 3% of their body weight. To maintain the homogeneity of EHP in the tank and minimize water exchange, a cloth-fiber filter was

implemented in each tank to recirculate the rearing water while removing waste. The water circulation was set up so that the water flow rate was 4500 L/hour using the water pump (SOBO WP-6000, SOBO Company, China). Water quality was monitored to maintain optimal levels throughout the experiment at 28–30°C, salinity levels at 20 ppt, DO > 4 mg/L (Pro 20 Dissolved Oxygen Meter, YSI, USA), and pH 7.8–8.3 (PARA pH test, Aquacare 2000.2, England). Total ammonia nitrogen < 5 ppm and nitrite < 0.5 ppm were monitored using the Para Ammonia Test (Aquacare 2000.4, England) and the SONA Nitrite Test (PARA Test, Thailand), respectively.

## Methods of anesthesia and/or analgesia

Forty shrimp were collected from each tank on days 7 and 21 of the cohabitation. A 2-step euthanasia procedure was performed following the AVMA Guidelines for the Euthanasia of Animals: 2013 Edition. For anesthesia, shrimp were immersed in 500 µL/L of Eugenol solution [25], followed by submerging in ice slurry for 15 min until there was no movement on the appendages. Shrimp hepatopancreases, stomachs, and intestines were dissected, flash frozen in liquid $N_2$, and stored at −80°C for further analysis. The decontamination process included treating shrimp rearing tanks and cultured water with 10% sodium hypochlorite and 50 ppm iodine for 2 hours. A diagram illustrating the EHP challenge and sample collection is provided in Fig 1.

## Quantitative real-time PCR analysis to determine EHP infection levels in shrimp hepatopancreas

The level of EHP infection in shrimp hepatopancreas was determined using the SWP-PCR [3]. The forward primer sequence (SWP-1F) was 5' TTGCAGAGTGTTGTTAAGGGTTT 3' and the reverse primer sequence (SWP-2R) was 5' GCTGTTTGTCTCCAACTGTATTTGA 3'. The qPCR reaction (20 µL each) contained 1X SYBR Green PCR Master mix,

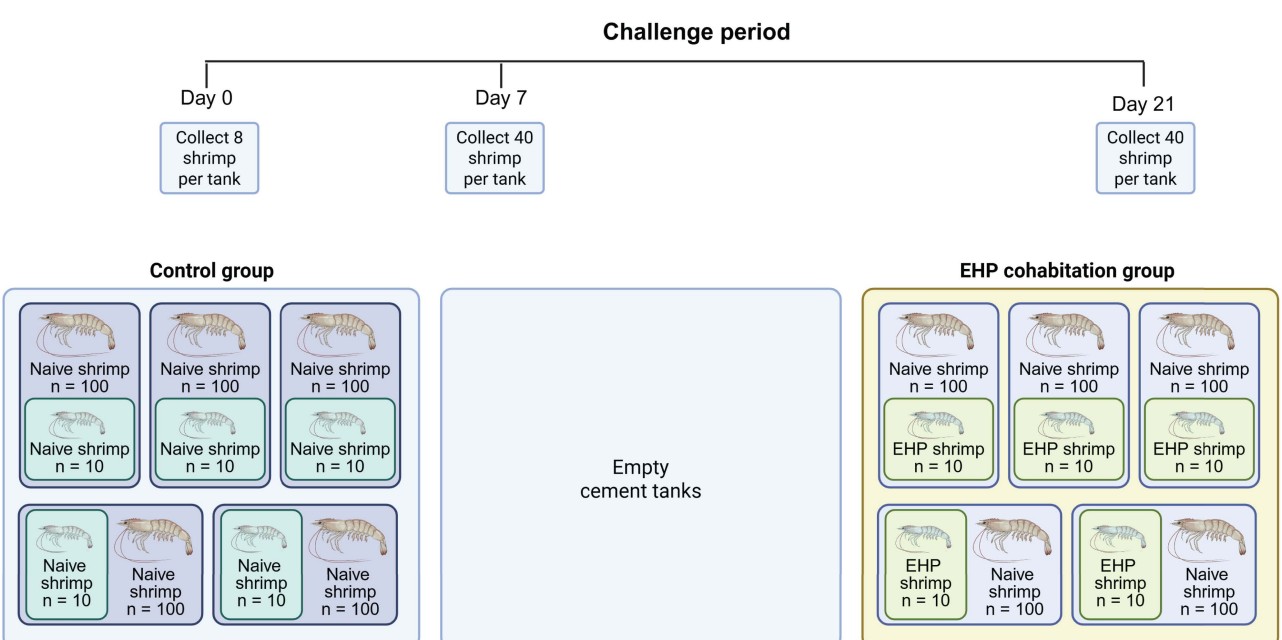

**Fig 1. Outline for the EHP cohabitation challenge.** Juvenile *L. vannamei* at 3 g body weight were transferred to ten 600-liter tanks (*n* = 100 in each tank) and acclimated for 24 hours. On day 0, a smaller plastic basket containing either 10 naïve shrimp or 10 EHP-infected shrimp was submerged inside the 600-liter tank to obtain the control or the EHP-infected shrimp, respectively. Shrimp hepatopancreases, stomachs, and intestines were collected from each tank on days 0, 7, and 21. Three shrimp from each rearing tank were fixed with paraformaldehyde for immunohistochemical analysis. The remaining samples were flash-frozen in liquid $N_2$ and stored at −80°C.

10 ng of total DNA extract from shrimp hepatopancreas, 0.2 μM of SWP-1F primer, and 0.2 μM of SWP-2R primer. The qPCR conditions included 1 cycle at 95°C for 15 s, 40 cycles of 95°C for 15 s, 64°C for 30 s, and 72°C for 30 s, and a dissociation stage at 95°C for 15 s, 60°C for 1 min, 95°C for 15 s, and 60°C for 1 min. A standard curve was constructed using a standard EHP-SWP plasmid with a 10-fold dilution that resulted in plasmid concentrations ranging from 0 to $10^6$ copies/μL. The qPCR analysis was performed using ABI Prism 7500 Sequence detection software (AB Applied Biosystems, Easter City, California, USA). The amplification efficiency was 85.4% and 86.6% for the qPCR reaction to monitor samples collected on days 7 and 21, respectively. The $R^2$ of the standard curve was 0.999 for both analyses.

### Ethyl acetate extraction of eicosanoids in shrimp tissues

Shrimp organs were homogenized in liquid $N_2$ using a pestle and mortar. The tissues were dissolved in Hank's Balanced Salt Solution (Invitrogen, California, USA) and homogenized again using a plastic pestle until no tissue residue was visible. The tissue concentrations were adjusted to 0.2 g/mL for hepatopancreases and 0.1 g/mL for stomachs and intestines. Five microliters of glacial acetic acid were added to each aliquot to adjust the tissue homogenate to pH 4.0. Ten microliters of 10% butylated hydroxytoluene in ethanol (*w/v*) were added along with three internal standards, prostaglandin $E_2$-$d_4$, 5(S)-HETE-$d_8$, and EPA-$d_5$. Ethyl acetate extraction was performed using a 1:1 ratio (*v/v*) between tissue homogenates and ethyl acetate, as previously described [26]. The resulting extracts were dried and dissolved in 100% ethanol for the ultra-high performance liquid chromatography tandem mass spectrometry (UHPLC-HRMS/MS) analysis.

### UHPLC-HRMS/MS analysis of eicosanoids in shrimp organs

UHPLC-HRMS/MS analysis was conducted using a DIONEX 3000 RS UHPLC system coupled with an Orbitrap Fusion™ Tribrid™ mass spectrometer. The system was operated with Xcalibur software (version 4.4.16.14) and calibrated using the Pierce™ FlexMix™ calibration solution (Thermo Fisher Scientific, USA). Chromatographic separation was performed on a Waters Acclaim™ RSLC 120 C18 (2.1 × 150 mm, 2.2 μm). The mobile phase consisted of 0.01% (*v/v*) acetic acid in water and 0.01% (*v/v*) acetic acid in acetonitrile. The elution program and mass spectrometer parameters were set as described in Yotbuntueng et al., 2022 [26].

### Transcriptional analysis of eicosanoid biosynthesis genes

Tissue homogenization and RNA extraction were performed in RNase-free conditions. Total RNA was obtained from shrimp hepatopancreases, stomachs, and intestines using the Trizol reagent (Invitrogen) and treated with DNase (Promega, Wisconsin, USA). The cDNA synthesis was performed on the resulting RNA using the RevertAid™ First Strand cDNA Synthesis Kit (Thermo Fisher Scientific, Massachusetts, USA).

The qPCR analysis was performed using the SsoFast™ EvaGreen® Supermix (Bio-Rad, California, USA). Fifty to two hundred nanograms of cDNA were used as templates for each real-time PCR reaction. The transcription levels of the gene of interest relative to those of the housekeeping gene *L. vannamei elongation factor 1α* (*LvEF1α*) were obtained using the standard curve method [27]. Primers sequences and PCR conditions are provided in S1 Table.

### Immunohistochemical analysis of shrimp hepatopancreas

A total of 30 shrimp cephalothoraxes from the EHP cohabitation experiment (15 from each group) were fixed with Davidson's AFA (22% formalin, 11.5% acetic acid, 31.5% ethanol in distilled water) for 24 hours [28]. The Davidson's AFA was subsequently removed and replaced with an equal volume of 70% ethanol. The tissue samples were subjected to dehydration, clearing, and paraffin infiltration in an automatic tissue processor (Leica TP 1020). The shrimp tissues were embedded in paraffin blocks, sectioned at 5 μm thickness, and placed on the Histogrip (Invitrogen) coated glass slides.

The obtained tissue sections were incubated in 1% hydrogen peroxide in 70% ethanol for 30 min to remove the paraffin. The deparaffinized section was subsequently rehydrated with 1% glycine in PBS for 10 min. The antigen retrieval process was performed by incubating the slides in sodium citrate buffer (pH 6.0) at 98°C for 20 min. The slides were incubated with 0.5% Triton X-100 in PBS for 8 min and washed twice with PBS for 5 min. The slides were then incubated with a blocking buffer (10% normal goat serum and 2% BSA in PBS) for 1 hour. Two antibodies were used for this analysis: the anti-cyclooxygenase-1 (COX-1) antibody (Cat No. ab244261, AbCAM, United Kingdom) and the anti-*L. vannamei* prostaglandin F synthase (LvPGFS) antibody (Cat No. SC2039-PF, Genscript, Singapore). The anti-COX-1 antibody is a commercially available polyclonal antibody raised against human COX-1. The anti-LvPGFS antibody is a custom-made polyclonal antibody raised against a full-length recombinant *L. vannamei* PGFS. Each antibody was diluted at 1:100 (*v/v*) and added to separate tissue sections. The sections were then incubated overnight in a moist chamber at 37°C. The unbound antibody was washed in PBS three times, 5 min each. The sections were exposed to a goat-anti-rabbit IgG conjugated with horseradish peroxidase at a 1:500 dilution (*v/v*) for 1 hour at room temperature and washed with PBS three times for 5 min each.

The expression of the *L. vannamei* COX (LvCOX) and LvPGFS in the tissue sections was detected by the reaction of peroxidase with a peroxidase substrate kit (VECTOR NovaRED, USA) for 5 min. The tissues were washed with distilled water and counterstained with hematoxylin for 30 s. Subsequently, the tissues were dehydrated, cleared, and mounted. The stained tissues were analyzed and photographed [29]. The captured micrographs were analyzed using the ImageJ program (LOCI, University of Wisconsin) [30].

### Statistical analysis

The *t*-test was performed to determine significant differences between the means of independent samples with the threshold for significance at $P < 0.05$ (*) or $P < 0.01$ (**). Dunnett's test was also performed to compare samples of the same condition to those collected on day 0 (†, # for $P < 0.05$ and ††, ## for $P < 0.01$).

## Results

### Determining EHP copy number using the SWP-qPCR analysis

The shrimp hepatopancreases collected from the cohabitation experiment were subjected to DNA extraction and qPCR analysis using the *SWP* primers. All control samples were tested negative for EHP infection (Fig 2). The qPCR analysis revealed that the average EHP copy number in shrimp hepatopancreases collected on day 7 from tanks 1, 2, and 3 ranged from 217.12–3,633.37 copies/ng DNA. However, the qPCR analysis failed to detect the *SWP* gene in tanks 4 and 5 (S1 File), suggesting that the EHP copy numbers in these tanks were below the limit of detection. As a result, data obtained from the EHP cohabitation samples collected on day 7 from tanks 4 and 5 were excluded from further analysis. On the other hand, shrimp samples collected on day 21 showed the average EHP copy numbers between 5,104.50–9,874.96 copies/ng DNA (Fig 2).

### Analysis of eicosanoids in the shrimp hepatopancreas

Shrimp hepatopancreases collected from the cohabitation experiment were subjected to homogenization, ethyl acetate extraction, and UHPLC-HRMS/MS analysis. Thirteen eicosanoids, including two prostaglandins, five HETEs, and six HEPEs, were detected in the hepatopancreases of control and EHP-infected shrimp. The UHPLC-HRMS/MS analysis revealed that EHP infection altered the levels of eicosanoids in shrimp hepatopancreases differently in samples collected on days 7 and 21 of the cohabitation experiment. There was no significant change in the levels of eicosanoids between the control and EHP-infected group collected on day 7 (Fig 3). However, the hepatopancreases of EHP-infected shrimp collected on day 21 contained higher levels of $PGF_{2\alpha}$ and 15-deoxy-$\Delta^{12,14}$-prostaglandin

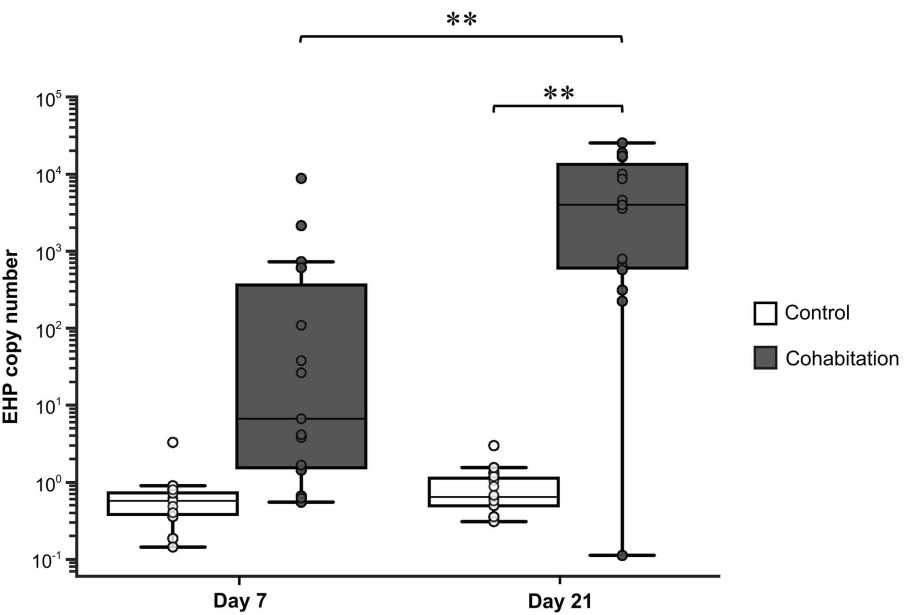

**Fig 2. EHP copy number per ng of shrimp hepatopancreas DNA obtained from the EHP cohabitation experiment.** Shrimp hepatopancreases from the control and cohabitation groups ($n=3$ per tank, 5 tanks per condition) were collected on days 7 and 21 for the analysis. The top and bottom of the box plot indicate data in quartiles 1 and 3, respectively. The line in the middle of the box plot indicates the median value. The top and bottom whiskers represent the minimum and maximum values of the EHP copy number, respectively. Each circle represents the EHP copy number from an individual shrimp. Asterisks indicate that the EHP copy numbers were significantly different between the two sets of shrimp samples using the $t$-test (* for $P<0.05$ and ** for $P<0.01$).

J$_2$ (15d-PGJ$_2$) but lower levels of 5-HETE, 9-HETE, 5-HEPE, 9-HEPE, and 18-HEPE than the control group (Fig 3). It should be noted that the levels of 12-HETE, which was relatively high in the control shrimp collected on day 0, decreased significantly in the control and EHP-infected shrimp collected on days 7 and 21 (Fig 3G). Additionally, the levels of 5-HEPE, 9-HEPE, and 18-HEPE also decreased in the control and EHP-infected shrimp collected on day 21 compared to those collected on day 0.

### Analysis of eicosanoids in the shrimp stomach

The UPHLC-HRMS/MS analysis of shrimp stomachs revealed the presence of 14 eicosanoids. Prostaglandin D$_2$ (PGD$_2$), which is a precursor of 15d-PGJ$_2$, was detected in shrimp stomachs along with the same 13 eicosanoids previously identified in shrimp hepatopancreases. On day 7, the levels of 12-HETE and 12-HEPE (Fig 4H and 4L) increased in the stomachs of EHP-infected shrimp compared to those of the control group. However, changes in the eicosanoid profiles became more apparent in the samples collected on day 21. The levels of PGD$_2$, PGF$_{2\alpha}$, 15d-PGJ$_2$, 5-HETE, 8-HETE, 9-HETE, 11-HETE, 12-HETE, 8-HEPE, 9-HEPE, 12-HEPE, and 15-HEPE were higher in the stomachs of EHP-infected shrimp compared to the control shrimp (Fig 4). When the shrimp samples were compared between different days of the experiment, it was observed that the levels of 5-HETE, 9-HETE, 5-HEPE, 9-HEPE, 15-HEPE, and 18-HEPE were significantly lower in the stomachs of both the control and EHP-infected shrimp collected on days 7 and 21 compared to day 0 (Fig 4D, 4F, 4I, 4K, 4M, and 4N). On the other hand, the levels of PGF$_{2\alpha}$, 15d-PGJ$_2$, and 12-HETE in control and EHP-infected shrimp on day 21 were higher than the control samples collected on day 0 (Fig 4B, 4C, and 4H). This suggests that other factors, such as the water temperature, molting period, or stress, leads to additional upregulation of eicosanoids in the control samples on day 21 of the cohabitation experiment.

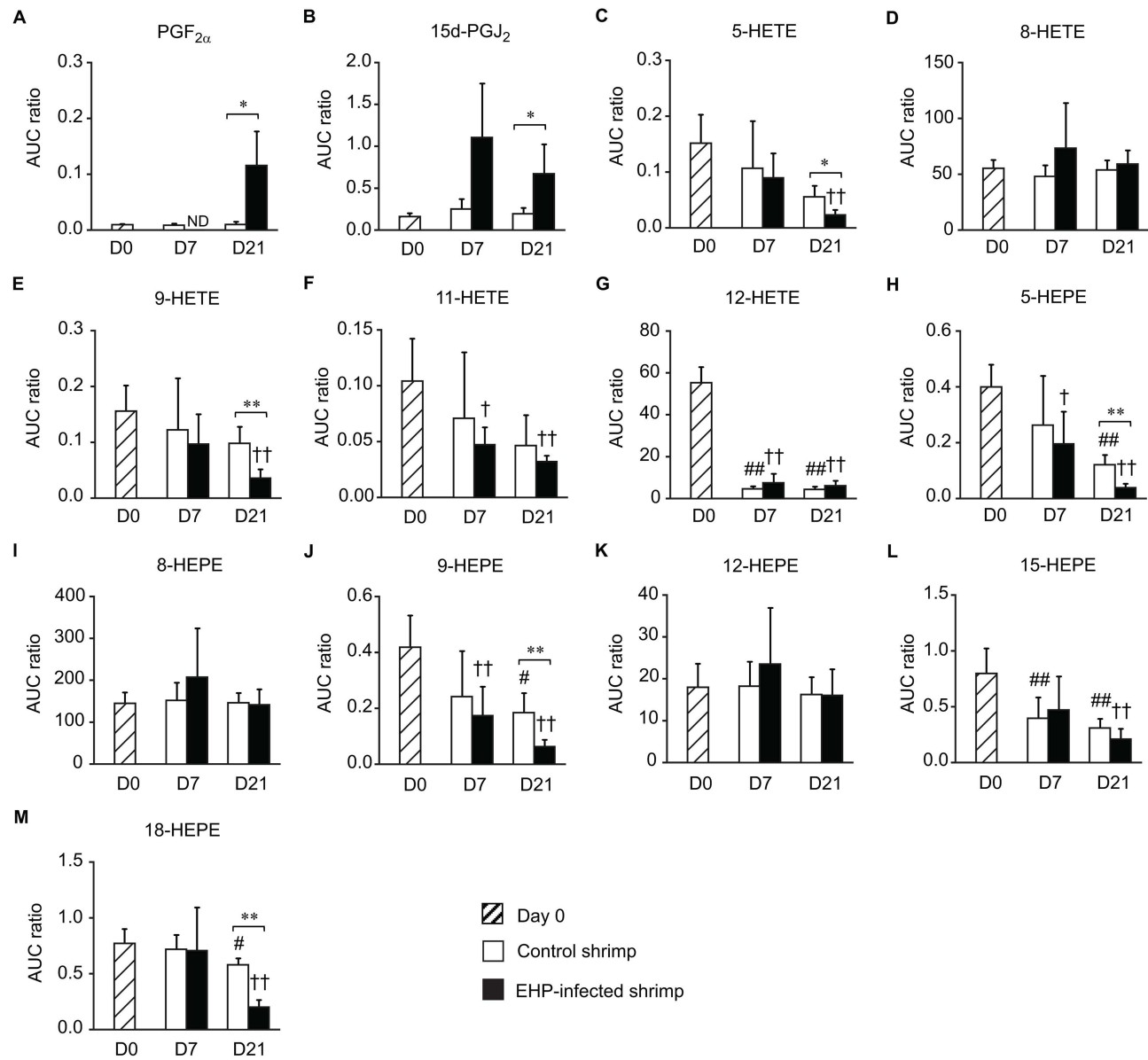

**Fig 3. The levels of eicosanoids in the hepatopancreases of control and EHP-infected shrimp.** The relative abundance of eicosanoids in shrimp hepatopancreases was determined using the ratios between the area under the curve (AUC) of the target eicosanoid and the internal standard 12(S)-HETE-$d_8$. The relative abundance of (A) PGF$_{2\alpha}$, (B) 15d-PGJ$_2$, (C) 5-HETE, (D) 8-HETE, (E) 9-HETE, (F) 11-HETE, (G) 12-HETE, (H) 5-HEPE, (I) 8-HEPE, (J) 9-HEPE, (K) 12-HEPE, (L) 15-HEPE, and (M) 18-HEPE were compared between the hepatopancreases of control (white bars) and EHP-infected shrimp (black bars) using the $t$-test (* for $P < 0.05$ and ** for $P < 0.01$). Striped bars indicate shrimp samples collected on day 0. Data is shown as means ± SD. Hashes indicate significant differences in the eicosanoid levels in the hepatopancreases of the control group collected on day 0 and those collected on days 7 and 21 using Dunnett's test (# for $P < 0.05$ and ## for $P < 0.01$). Crosses indicate significant differences in the eicosanoid levels in the hepatopancreases of EHP-infected shrimp collected on day 0 and those collected on days 7 and 21 using Dunnett's test († for $P < 0.05$ and †† for $P < 0.01$).

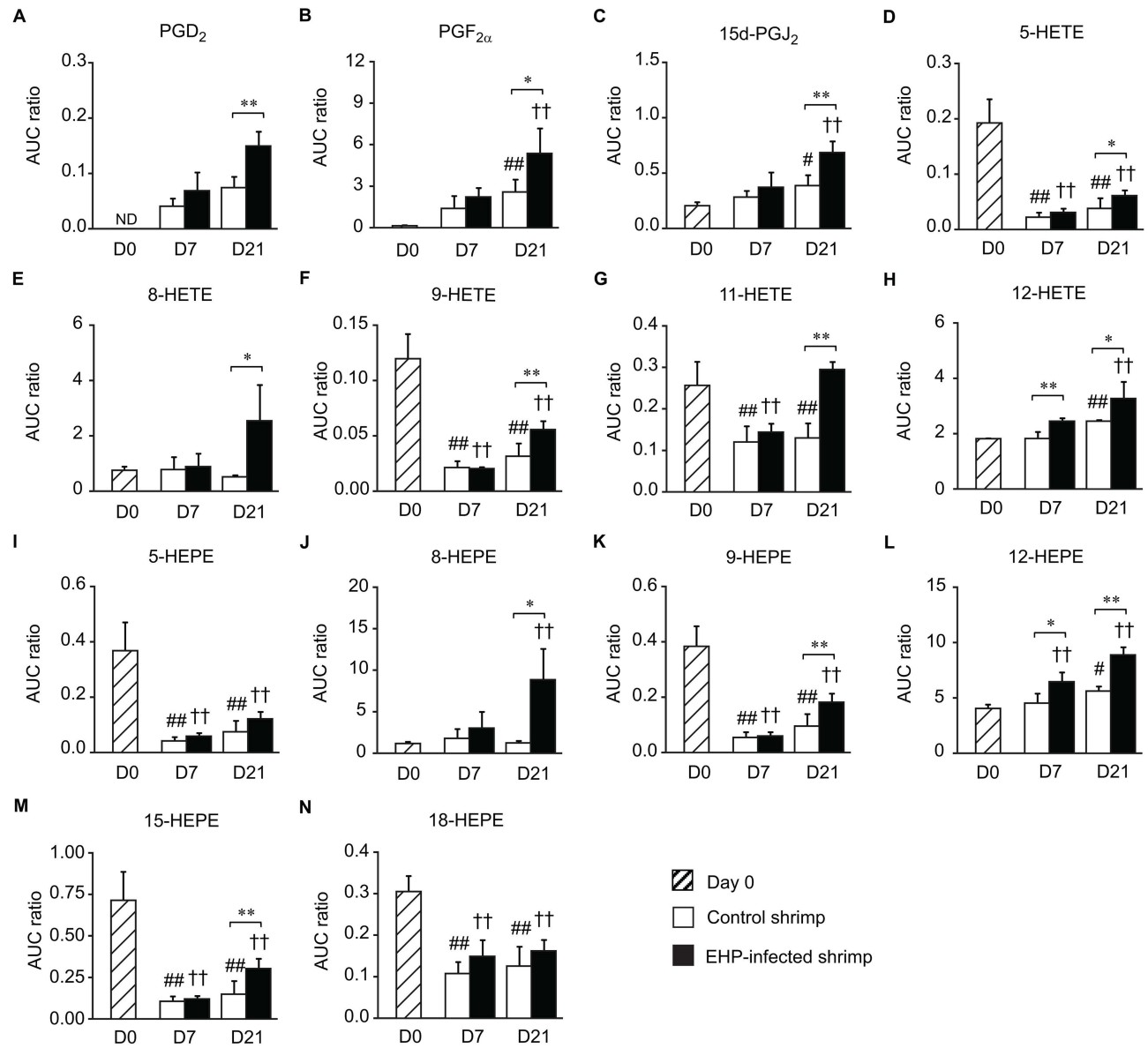

**Fig 4. The levels of eicosanoids in the stomachs of control and EHP-infected shrimp.** The relative abundance of (A) $PGD_2$, (B) $PGF_{2\alpha}$, (C) 15d-$PGJ_2$, (D) 5-HETE, (E) 8-HETE, (F) 9-HETE, (G) 11-HETE, (H) 12-HETE, (I) 5-HEPE, (J) 8-HEPE, (K) 9-HEPE, (L) 12-HEPE, (M) 15-HEPE, and (N) 18-HEPE were compared between the stomachs of control (white bars) and EHP-infected shrimp (black bars). Striped bars indicate shrimp samples collected on day 0. Data are shown as means ± SD. Asterisks indicate statistically significant differences in the eicosanoid levels in the stomachs of control and EHP-infected shrimp using the $t$-test (* for $P<0.05$ and ** for $P<0.01$). Hashes indicate significant differences in the eicosanoid levels in the stomachs of control shrimp on day 0 compared to those on days 7 and 21 using Dunnett's test (# for $P<0.05$ and ## for $P<0.01$). Crosses indicate significant differences in the eicosanoid levels in the stomachs of EHP-infected shrimp collected on day 0 compared to those on days 7 and 21 using Dunnett's test († for $P<0.05$ and †† for $P<0.01$).

## Analysis of eicosanoids in the shrimp intestine

The UHPLC-HRMS/MS analysis of shrimp intestinal extracts revealed the presence of the same 14 eicosanoids previously identified in shrimp stomachs. While minimal changes were observed in the hepatopancreases and stomachs collected on day 7 (Figs 3 and 4), the levels of eicosanoids in the shrimp intestines were severely affected by EHP

infection. The intestines of EHP-infected shrimp contained higher levels of $PGF_{2\alpha}$, 8-HETE, 5-HEPE, 8-HEPE, 9-HEPE, and 12-HEPE than the control group (Fig 5). When shrimp samples were collected on day 7, the surge in the levels of eicosanoids subsided, resulting in comparable levels of eicosanoids between the control and EHP-infected groups. A

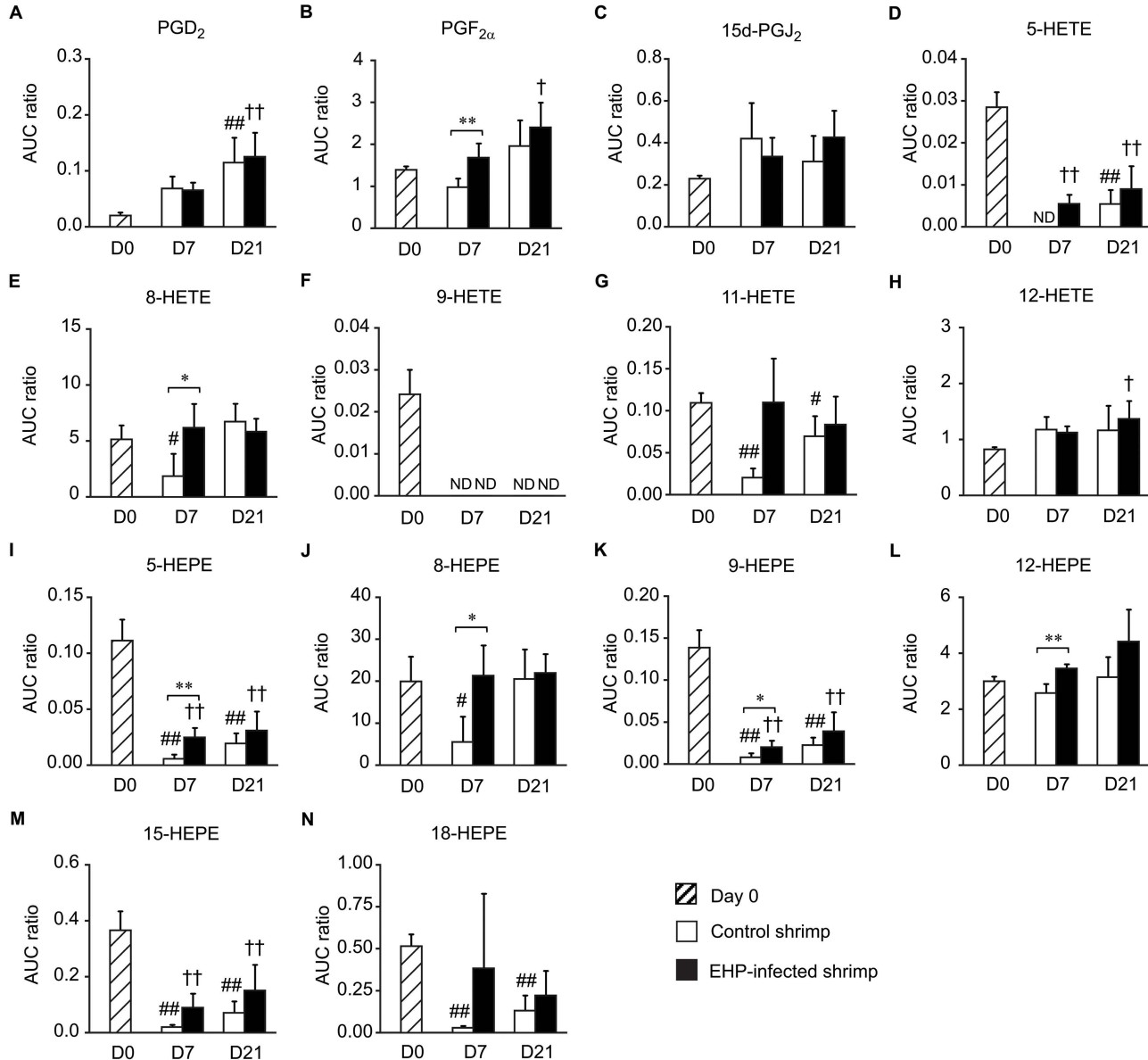

**Fig 5. The levels of eicosanoids in the intestines of control and EHP-infected shrimp.** The relative abundance of (A) $PGD_2$, (B) $PGF_{2\alpha}$, (C) 15d-$PGJ_2$, (D) 5-HETE, (E) 8-HETE, (F) 9-HETE, (G) 11-HETE, (H) 12-HETE, (I) 5-HEPE, (J) 8-HEPE, (K) 9-HEPE, (L) 12-HEPE, (M) 15-HEPE, and (N) 18-HEPE were compared between the intestines of control (white bars) and EHP-infected shrimp (black bars). Striped bars indicate shrimp samples collected on day 0. Data are shown as means±SD. Asterisks indicate statistically significant differences in the eicosanoid levels in the intestines of control and EHP-infected shrimp using the $t$-test (* for $P<0.05$ and ** for $P<0.01$). Hashes indicate significant differences in the eicosanoid levels in the intestines of control shrimp on day 0 compared to days 7 and 21 using Dunnett's test (# for $P<0.05$ and ## for $P<0.01$). Crosses indicate significant differences in the eicosanoid levels in the intestines of EHP-infected shrimp on day 0 compared to those on days 7 and 21 using Dunnett's test († for $P<0.05$ and †† for $P<0.01$).

comparative analysis was performed on the levels of eicosanoids from days 0, 7, and 21, revealing that the levels of 5-HETE, 9-HETE, 5-HEPE, 9-HEPE, and 15-HEPE in the intestines of shrimp collected on day 0 were higher than those of the control and EHP-infected shrimp collected on days 7 and 21 (Fig 5D, 5F, 5I, 5K, and 5M). On the other hand, the levels of $PGD_2$ in the samples collected on day 0 were lower than those of the control and EHP-infected groups collected on day 21 (Fig 5A). This data indicates that EHP infection is not the only contributing factor for altering the eicosanoid biosynthesis pathway in the shrimp gastrointestinal tract.

## Immunohistochemical analysis of COX and PGFS in shrimp hepatopancreas

Immunohistochemical analysis was performed to determine the amount and localization of two prostaglandin biosynthesis enzymes, LvCOX and LvPGFS, in shrimp hepatopancreases. The LvCOX, which regulates the rate-limiting step in prostaglandin biosynthesis, was detected using a COX-1 antibody. The red-brown precipitates were localized in the apical compartment of B cells in both control and EHP-infected shrimp on days 7 and 21 (Fig 6A–6D, black arrows). These LvCOX immunoreactive signals were subjected to semi-quantitative analysis using the ImageJ software. The signals from the EHP-infected shrimp were higher than those of the control shrimp on days 7 and 21 of the cohabitation experiment (Fig 6E).

The localization of LvPGFS, which catalyzes the biosynthesis of $PGF_{2\alpha}$, was also detected as red-brown precipitates in the apical compartment of B cells (Fig 7A–7D; black arrows) and E cells (Fig 7A–7D; white arrows) in the hepatopancreases of the control and the EHP-infected shrimp. A semi-quantitative analysis revealed that the LvPGFS immunoreactive signals were higher in the hepatopancreases of EHP-infected shrimp than in the control shrimp collected on days 7 and 21 (Fig 7E).

## Transcriptional analysis of prostaglandin biosynthesis genes in the shrimp gastrointestinal tract

The transcription levels of *LvcPLA2, LvCOX,* and *LvPGFS* were determined in shrimp hepatopancreases, stomachs, and intestines collected from the EHP cohabitation experiment. Although EHP infection altered the metabolic profiles in these organs, little impact was observed on the transcription levels of eicosanoid biosynthesis genes (Fig 8). The levels of *LvcPLA2, LvCOX,* and *LvPGFS* transcripts were comparable in shrimp hepatopancreases collected on days 0, 7, and 21 of the cohabitation experiment (Fig 8A–8C). In the shrimp stomachs, the transcription levels of *LvcPLA2* were below the limit of detection on day 0 and in the control collected on day 21. Additionally, EHP infection resulted in the upregulation of *LvcPLA2* in the stomachs collected on day 21 when compared with the control samples (Fig 8D). On the other hand, there was no significant difference in the transcription levels of *LvCOX* and *LvPGFS* in shrimp stomachs throughout the cohabitation experiment (Fig 8E and 8F). In shrimp intestines, the levels of *LvcPLA2* transcripts were below the limit of detection on day 0 and in the EHP-infected shrimp collected on day 7 (Fig 8G). The upregulation of *LvcPLA2* in the control shrimp collected on day 7 was likely due to other factors, such as shrimp feed, molting period, or rearing environment. Additionally, the levels of *LvcPLA2* were also comparable between the control and EHP-infected shrimp collected on day 21, further confirming that the EHP infection had no impact on the levels of *LvcPLA2* in shrimp intestines. Similarly, the levels of *LvCOX* in shrimp intestines were comparable among all treatments collected at any time points (Fig 8H), suggesting that EHP infection does not affect the transcription levels of *LvCOX* in shrimp intestines. Lastly, the transcription levels of *LvPGFS* were reduced to below the limit of detection in EHP-infected shrimp collected on day 7, suggesting that EHP infection suppressed the production of *LvPGFS* transcripts (Fig 8I).

## Analysis of PUFAs in the shrimp gastrointestinal tract

The UHPLC-HRMS/MS analysis was performed to determine whether the increasing levels of eicosanoids were due to increased substrate availability. The levels of PUFAs were first compared between the control and EHP-infected shrimp

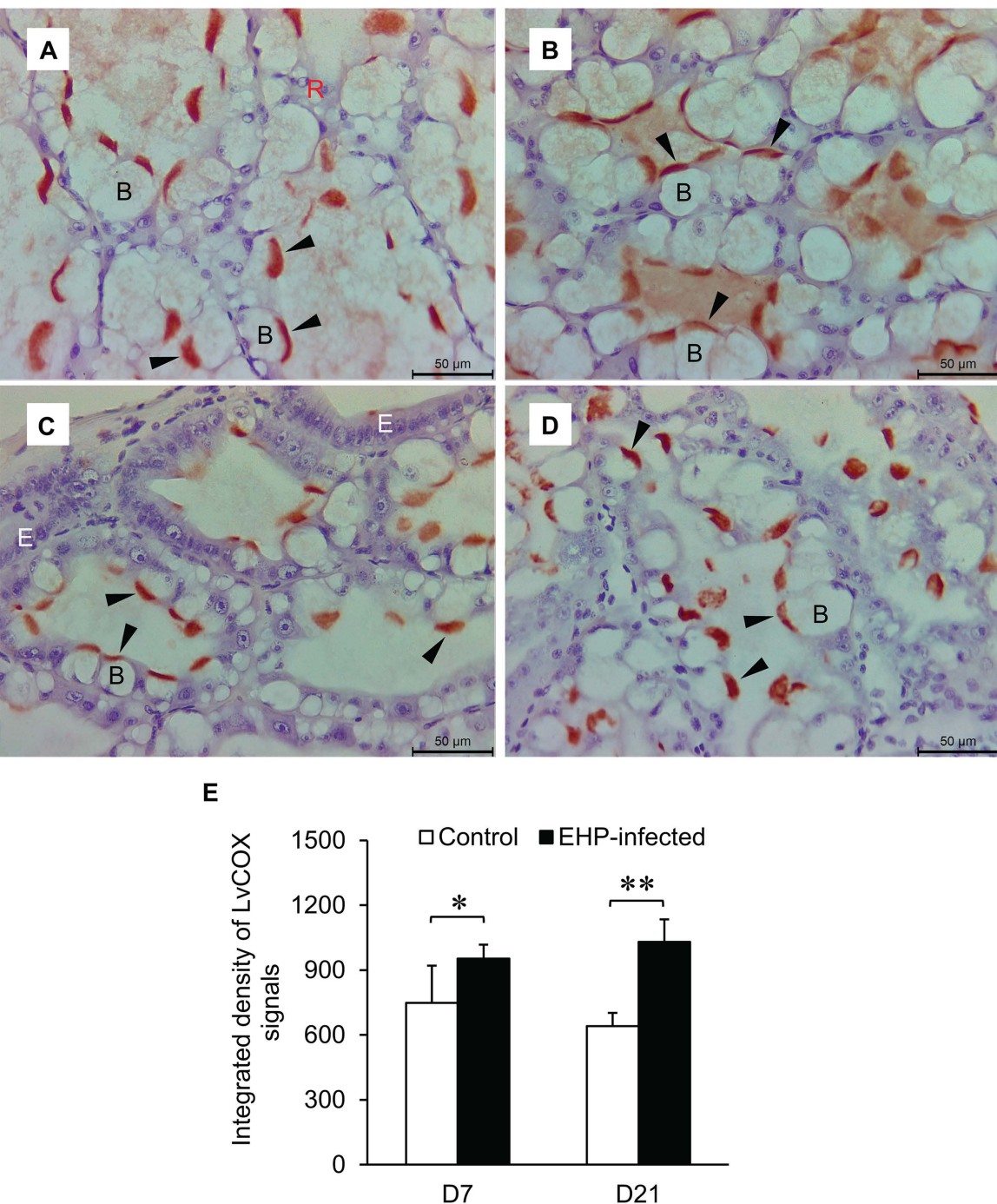

**Fig 6. Expression and localization of LvCOX in shrimp hepatopancreases.** Shrimp from the control (A and C) and EHP-infected (B and D) groups were collected on days 7 (A and B) and 21 (C and D). Shrimp hepatopancreases were subjected to paraffin embedding, H&E staining, and immunohisto-chemistry staining using a commercially available polyclonal antibody against mammalian COX-1. The black arrowhead indicates the localization of the immunoreactive signal to the COX-1 antibody at the apical compartment of B cells. Letters B (black), E (white), and R (red) are used to label the B, E, and R cells, respectively. (E) A semi-quantitative analysis of the COX-1 immunoreactive signals was performed on the control and EHP-infected groups on days 7 and 21 (* for $P < 0.05$ and ** for $P < 0.01$).

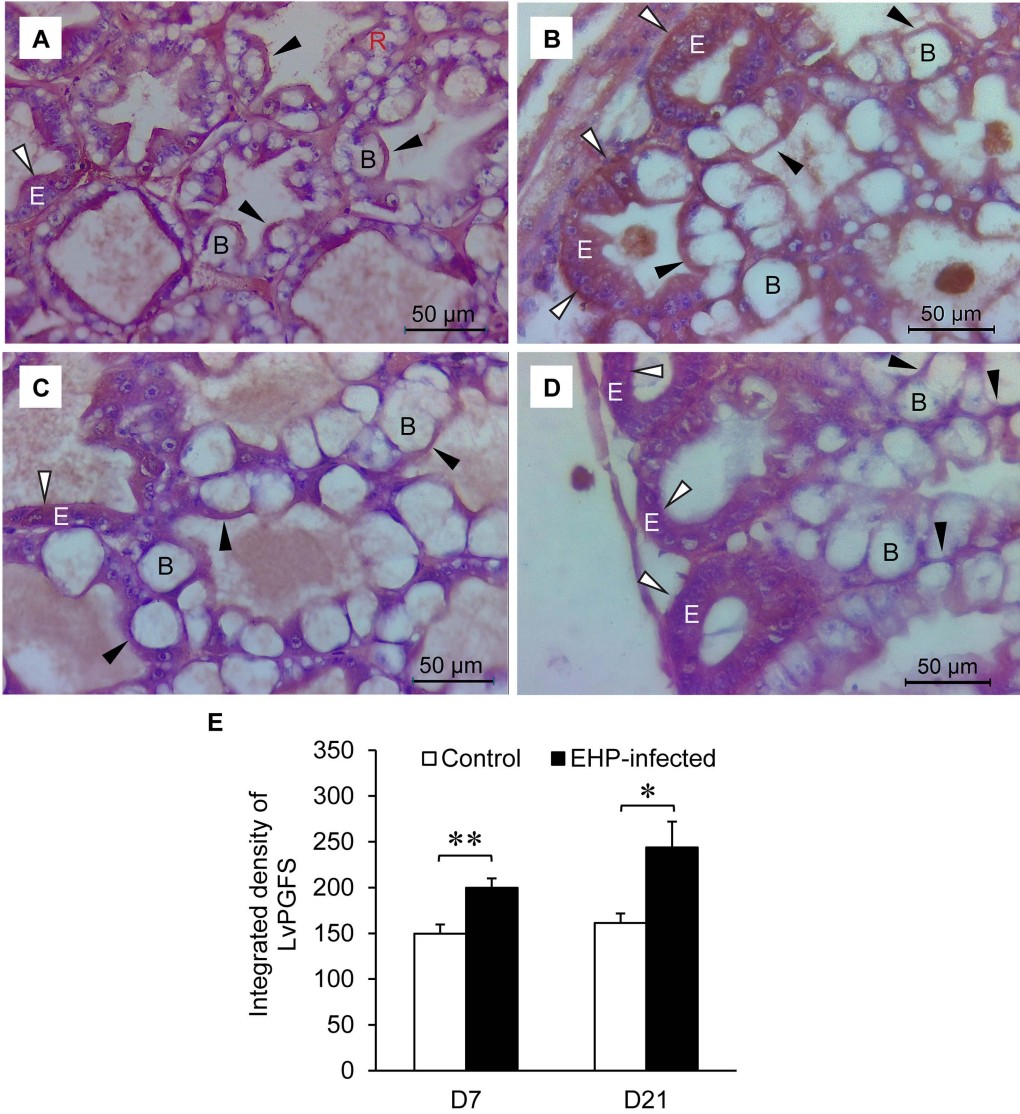

**Fig 7. Expression and localization of LvPGFS in shrimp hepatopancreases.** Shrimp hepatopancreases were subjected to paraffin embedding, H&E staining, and immunohistochemistry staining using antibodies against LvPGFS. Shrimp from the control (A and C) and EHP-infected (B and D) groups were collected on days 7 (A, B) and 21 (C, D). The red-brown precipitates indicated that LvPGFS was localized in the apical compartment of B cells (black arrow) and distal zones in E cells (white arrow). Letters B (black), E (white), and R (red) are used to label the B, E, and R cells, respectively. (E) A semi-quantitative analysis of the LvPGFS immunoreactive signal was performed on the control and EHP-infected shrimp collected on days 7 and 21. Significant differences in the LvPGFS immunoreactive signals were determined using the *t*-test (* for $P < 0.05$ and ** for $P < 0.01$).

collected at the same time point. The hepatopancreases and stomachs of EHP-infected shrimp collected on days 7 and 21 contained higher levels of ARA than the control shrimp (Fig 9A and 9C). Meanwhile, the ARA levels in shrimp intestines remained unchanged throughout the cohabitation experiment (Fig 9E). EHP infection also increased the levels of EPA in the hepatopancreases collected on day 21 (Fig 9B). Additionally, the EPA levels in the stomachs and intestines of EHP-infected shrimp collected on days 7 and 21 were higher than those of the control shrimp (Fig 9D and 9F).

To determine whether changes in the levels of PUFAs were due to other factors aside from EHP infection, the eicosanoid levels of the control shrimp collected on day 0 were compared with samples collected on days 7 and 21. In shrimp

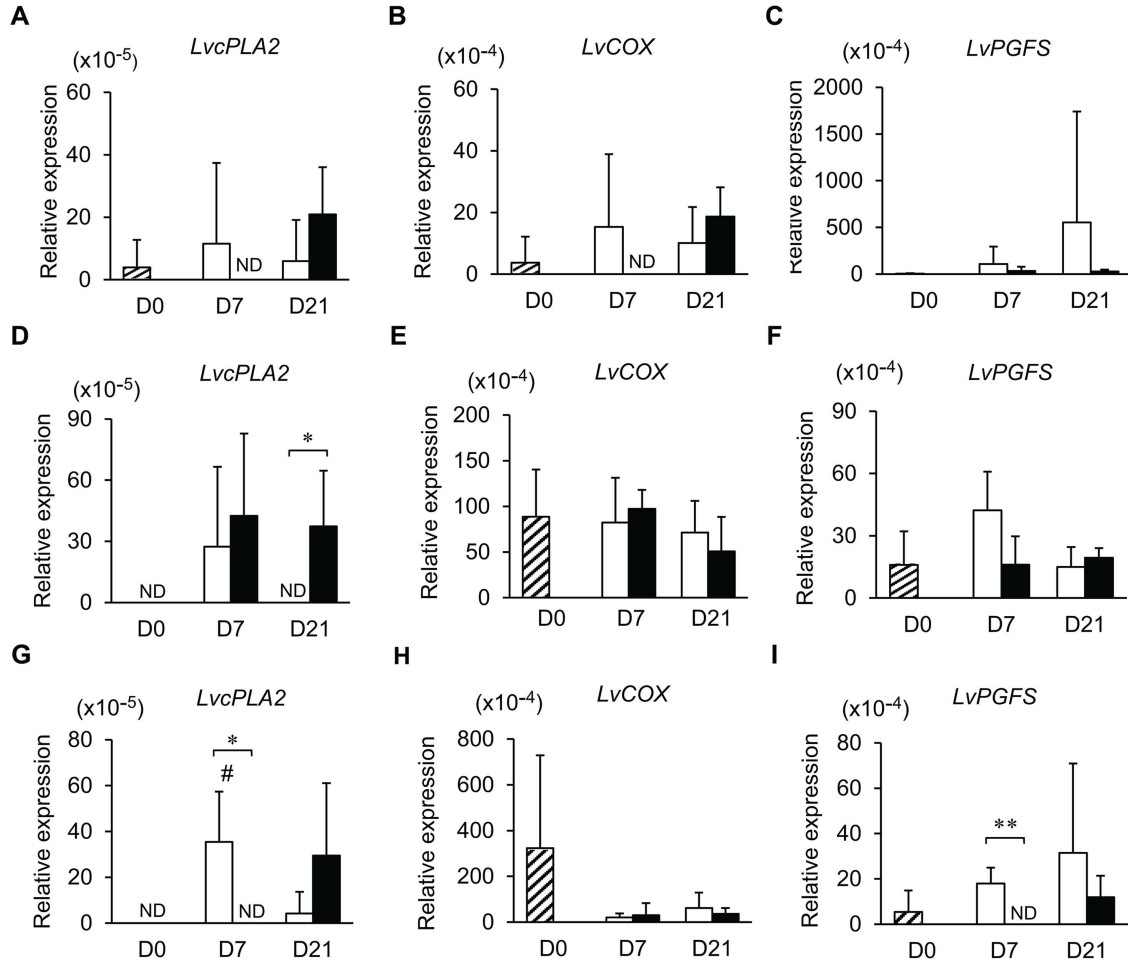

**Fig 8. The levels of prostaglandin biosynthesis gene transcripts in the gastrointestinal tract of control and EHP-infected.** Shrimp samples were collected on day 0 of the cohabitation experiment (striped bars) to establish the baseline. Shrimp from the control (white bars) and EHP-infected groups (black bars) were subsequently collected on days 7 and 21. The transcription levels of prostaglandin biosynthesis genes were examined in shrimp hepatopancreases (A to C), stomachs (D to F), and intestines (G to I). The transcription levels of *LvcPLA2* (A, D, and G), *LvCOX* (B, E, and H), and *LvPGFS* (C, F, and I) relative to *LvEF1α* were determined using the qPCR analysis. Asterisks indicate significant differences in the prostaglandin biosynthesis gene transcripts between the control and EHP-infected shrimp using the *t*-test (* for $P<0.05$ and ** for $P<0.01$). Hashes indicate that the transcription levels of prostaglandin biosynthesis genes collected on day 0 differed from the control shrimp collected on days 7 and 21 using Dunnett's test (# for $P<0.05$). The Dunnett's test was also performed on the samples collected on day 0 and the EHP-infected samples collected on days 7 and 21, but no significant difference was observed in the analysis.

hepatopancreas, the levels of ARA in the control shrimp collected on day 0 were lower than the EHP-infected shrimp collected on days 7 and 21 (Fig 9A). However, the levels of ARA increased significantly on day 21 in both the control and EHP-infected shrimp compared to those collected on day 0 (Fig 9A). Meanwhile, the levels of ARA in the stomachs and intestines of the control and EHP-infected shrimp collected on days 7 and 21 were comparable to those collected on day 0 (Fig 9C and 9E).

Changes in the levels of EPA were also examined between samples collected on days 0, 7, and 21. The levels of EPA in the stomachs of the control group collected on day 0 were lower than those of the EHP-infected group collected on day 21 (Fig 9D), which confirms that EHP infection elevated the levels of EPA in shrimp stomachs. On the other hand, there was a significant decrease in the levels of EPA in the intestines of the control shrimp collected on day 7 compared to those

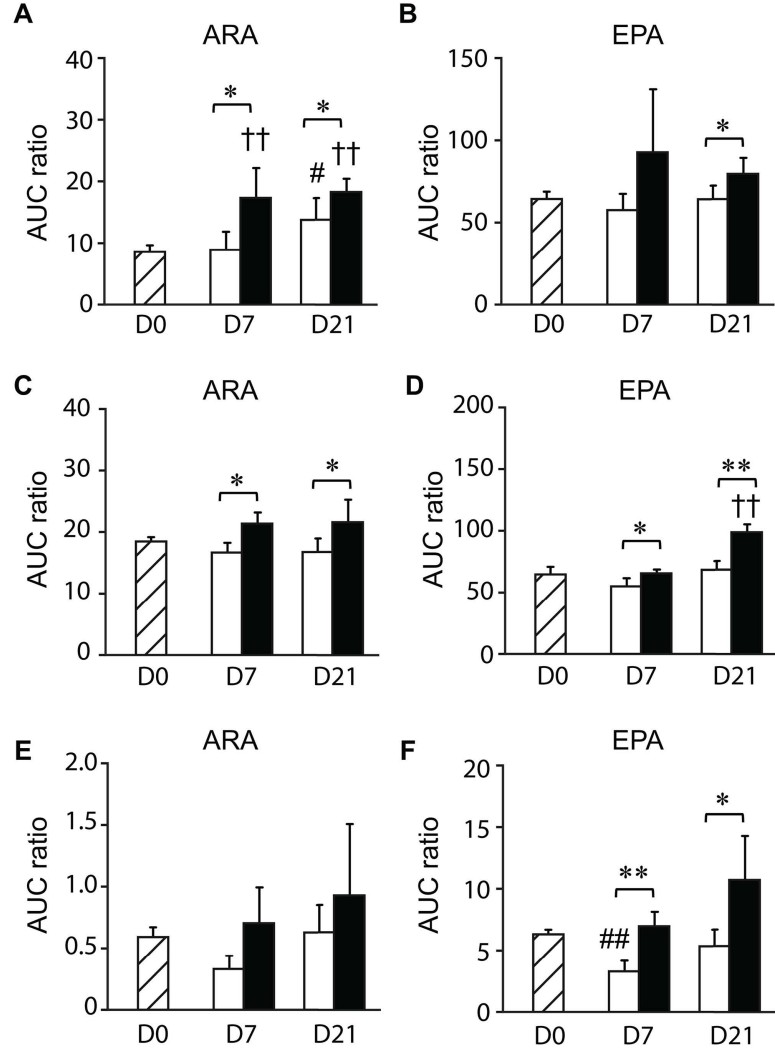

**Fig 9. Levels of ARA and EPA in the gastrointestinal tract of control and EHP-infected shrimp.** The UHPLC-HRMS/MS analysis was performed to determine the levels of ARA (A, C, and E) and EPA (B, D, and F) in hepatopancreases (A and B), stomachs (C and D), and intestines (E and F) of shrimp in the control (white bars) and EHP-infected groups (black bars) collected on days 7 and 21. Striped bars represent the control samples collected on day 0. Data is shown as means ± SD. Asterisks indicate statistically significant differences in the levels of PUFAs between the two conditions using the *t*-test (* for $P < 0.05$ and ** for $P < 0.01$). Hashes indicate statistically significant differences in the PUFA levels between the control shrimp collected on day 0 and those collected on days 7 and 21 using Dunnett's test (# for $P < 0.05$ and ## for $P < 0.01$). Crosses indicate statistically significant differences in the PUFA levels in the EHP-infected shrimp collected on day 0 and those collected on days 7 and 21 using Dunnett's test (†† for $P < 0.01$).

collected on day 0 (Fig 9F). This finding indicates that there are additional factors that contributed to the reduction of EPA levels in the intestines of shrimp in the control group collected on day 7.

## Discussion

This study examined changes in the eicosanoid biosynthesis pathway in the gastrointestinal tract of shrimp collected on days 0, 7, and 21 of the EHP cohabitation experiment. A combination of UHPLC-HRMS/MS and immunohistochemical analyses revealed that the eicosanoid biosynthesis pathway was upregulated at the enzymatic and metabolic levels in

the shrimp gastrointestinal tract. However, changes at the transcription levels of prostaglandin biosynthesis genes varied between organ types and infection time, which are summarized in Fig 10.

### EHP cohabitation challenge and SWP-qPCR detection

Several studies have published methods to induce EHP infection in *L. vannamei*, including ventral sinus injection of the purified EHP spores, oral gavage, reverse gavage, and cohabitation experiment [31–34]. Although the first four infection methods allow for the control of EHP spore numbers and ensure equal distribution of EHP spores among all the challenged shrimp, the cohabitation experiment was selected in this study due to 2 main reasons. First, the large number of shrimp samples required in this study makes it difficult to induce EHP infection via other methods. Second, the EHP cohabitation experiment was selected to mimic natural EHP infection in shrimp ponds. As a result, the levels of EHP infection in the cohabitated shrimp were unevenly distributed among samples within and across different challenge tanks, as observed in the SWP-qPCR in Fig 1. The qPCR analysis was unable to detect the SWP gene in shrimp hepatopancreases collected on day 7 from tanks 4 and 5 of the EHP cohabitation group. This may be due to (1) low levels of EHP-infection in shrimp samples from the commercial farm, (2) uneven distribution of EHP spores in the water circulation, and (3) low amplification efficiency for the SWP-qPCR analysis. The SWP-qPCR used in this study showed 85.4% and 86.6% amplification efficiencies (S1 Data). This could result in the under-representation of EHP copy number compared with the actual infection levels in the sampled shrimp. As the SWP copy number in the hepatopancreases of shrimp samples collected on day 7 from tanks 4 and 5 of the EHP-cohabitation groups was below the limit of detection, these samples were excluded from the analysis. Alternatively, the detection of polar tube protein 2 (PTP2) with the amplification efficiency at 102% would have provided a more accurate quantification of EHP copy number in shrimp samples in the future [35].

### The presence of EHP spores in the shrimp gastrointestinal tract and its impact on the shrimp's inflammatory response

Although EHP infection is primarily located in the shrimp hepatopancreas, several studies have demonstrated that other organs in the shrimp gastrointestinal tract, including the shrimp stomach and intestine, were also affected by EHP infection [7–9]. The presence of EHP spores in shrimp pancreatic tissues and feces has been demonstrated via Calcofluor

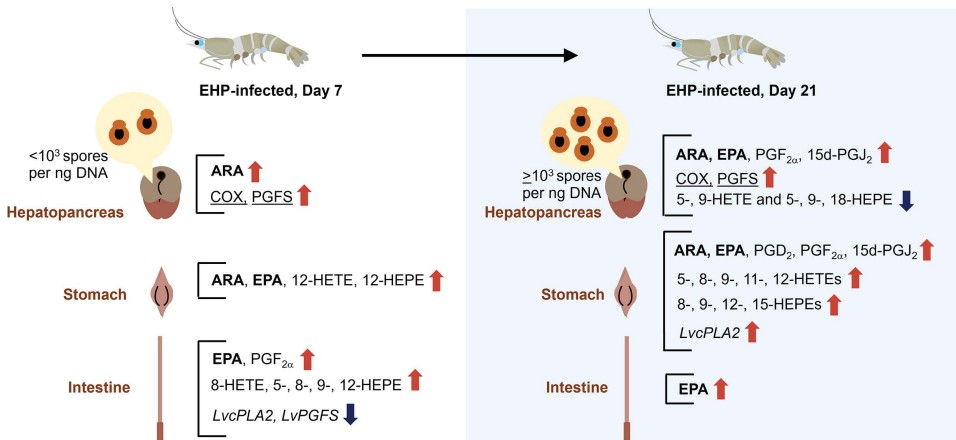

**Fig 10. Graphical representation of changes in the eicosanoid biosynthesis pathway in the shrimp gastrointestinal tract.** Changes in the levels of eicosanoids, PUFAs (bold), eicosanoid biosynthesis gene transcripts (italicized), and immunostaining signals of LvCOX and LvPGFS (underlined) were indicated using red and blue arrows, which signify the up- or down-regulation of the designated molecule when compared to the control group, respectively.

white staining [8]. Both germinated and non-germinated EHP spores were present in shrimp hepatopancreas and intestine, suggesting that spore germination can occur in both organs [7]. On the other hand, only the non-germinated spores were detected in the shrimp stomach, which excluded this organ from the possible EHP germination site [7]. Ni et al. 2025 performed histological examination of the stomach, hepatopancreas, and intestine of live, moribund, and dead shrimp that were infected with EHP [9]. The inflammatory responses were observed in the hepatopancreas and stomach of EHP-infected shrimp. Shrimp with high EHP copy number showed hematopoietic infiltration and the presence of inflammatory cells in the hepatopancreas and the pyloric stomach of [9]. This inflammatory response was also dose-dependent, as the hepatopancreatic tissues with high levels of EHP exhibited a more pronounced inflammatory response compared to those with low levels of EHP [9]. These findings concurred with the results from our UHPLC-HRMS/MS analysis. In samples collected on day 7, minimal changes were observed in the levels of eicosanoids in the shrimp stomach. However, changes in the eicosanoid profiles became more severe in samples collected on day 21, indicating that the inflammation in the shrimp stomach increased as the EHP infection progressed.

The levels of interleukin-6 (IL-6) and interleukin-8 (IL-8) were also measured in shrimp hepatopancreas using the commercially available enzyme immunoassay kits. However, neither IL-6 nor IL-8 was detected by the enzyme immunoassay (data not shown), suggesting that the antibodies raised against mammalian IL-6 and IL-8 failed to detect the IL-6 and IL-8 homologs in shrimp.

### Localization of prostaglandin biosynthesis enzymes in EHP-infected hepatopancreas

A histological section of EHP-infected hepatopancreas revealed that EHP spores localized in the cytoplasm of absorptive cells or R cells (nutrient storage cells), the secretory cells or B cells (storing and releasing digestive enzyme), and the fibrillar cells or F cells (synthesizing digestive enzyme) [6]. On the other hand, the embryonic cells or E cells in the distal region of the hepatopancreas remained uninfected [6]. In this study, the immunostaining signals revealed that LvCOX and LvPGFS were localized at the apical region of the B cells. Additionally, LvPGFS was detected in the E cells, suggesting that EHP infection resulted in the induction of eicosanoid biosynthesis in several cell types in the shrimp hepatopancreas. Moreover, the presence of LvCOX and LvPGFS indicates that B and E cells are the culprits for producing prostaglandins in the shrimp hepatopancreas.

### Pro- and anti-inflammatory roles of eicosanoids in the shrimp gastrointestinal tract

Amongst all the collected samples, EHP infection caused the most significant impact on the eicosanoid profiles in the shrimp intestine collected on day 7. The levels of $PGF_{2\alpha}$, 8-HETE, 5-HEPE, 8-HEPE, 9-HEPE, and 12-HEPE in the intestine of EHP-infected shrimp were elevated compared to the control shrimp. These data concurred with a histological analysis of the shrimp midgut region, in which the epithelial cells became necrotic or detached from the basement membrane in the intestine of shrimp infected with EHP [9]. Similarly, microsporidia infection in humans, including *Encephalitozoon hellem* and *Encephalitozoon intestinalis*, impaired the integrity of the host epithelial barrier, which led to inflammatory bowel disease [36]. Therefore, inflammation in the intestinal epithelial cells is likely a common effect of microsporidia infection.

This study established that EHP infection resulted in both the up- and down-regulation of eicosanoids in the shrimp gastrointestinal tract based on the comparison of the levels of eicosanoids between the control and EHP-infected samples. However, the levels of certain eicosanoids and PUFAs in the control samples also changed from days 0, 7, and 21. A study in *P. monodon* showed that the levels of ARA and DHA in the hepatopancreas of broodstock fed with commercial feed pellets (control group) increased from week 0 to week 4 [37]. This result is similar to the data from Fig 9A, in which the levels of ARA in the control group collected on day 21 were higher than those collected on day 0. This was likely due to the accumulation of ARA received from shrimp feed. Additionally, molting cycles may also affect the levels

of prostaglandins and hydroxy fatty acids. The transcriptomic analysis of *L. vannamei* showed that the levels of *LvCOX* transcripts changed according to the molting stages [38]. Therefore, any changes that occurred in both the control and EHP-infected shrimp on days 7 or 21, such as the reduction in the levels of 12-HETE in the hepatopancreas, could be due to changes in the molting stage. The changes in molting stage may lead to reduce the transcription levels of lipoxygenase enzymes that suppressed the production of eicosanoids. The same reasoning can also be applied to the decreasing levels of 5-HETE, 9-HETE, 5-HEPE, 9-HEPE, 15-HEPE, and 18-HEPE in the shrimp stomach, and the levels of 9-HETE, 5-HEPE, 9-HEPE, and 15-HEPE in the intestines. Additionally, changes in the water temperature may be responsible for the changes in the eicosanoid levels based on a study in the blue mussel *Mytilus edulis*, which demonstrated that eicosanoids are involved in the regulation of thermal stress response [39].

In mammals, eicosanoids play a significant role in pro- and anti-inflammatory responses [40–43]. Prostaglandin $E_2$ ($PGE_2$), $PGF_{2\alpha}$, and $PGD_2$ regulate inflammation and host immune response via the production of cytokines [40,41]. On the other hand, 15-HETE, 11-HEPE, 12-HEPE, and 15-HEPE display anti-inflammatory properties [41–43]. In this study, the hepatopancreases of EHP-infected shrimp collected on day 7 showed higher immunostaining signals of LvCOX and LvPGFS than in the control shrimp. Additionally, the levels of $PGF_{2\alpha}$ in shrimp intestines collected on day 7 were also higher than the control shrimp. This suggests that the pro-inflammatory response was activated as soon as day 7 of the cohabitation experiment. As the EHP infection progressed to day 21, the levels of 5-HETE, 9-HETE, 5-HEPE, 9-HEPE, and 18-HEPE decreased in shrimp hepatopancreases. In mammals, $15d\text{-}PGJ_2$ was typically associated with an anti-inflammatory pathway. However, the increasing levels of $15d\text{-}PGJ_2$ in shrimp hepatopancreases matched the increasing levels of its substrate, ARA, and another pro-inflammatory signaling molecule, $PGF_{2\alpha}$, in shrimp hepatopancreases and stomachs. Studies have shown that $15d\text{-}PGJ_2$ serves a dual function as both a pro- and anti-inflammatory molecule [44–46]. In eosinophils, $15d\text{-}PGJ_2$ acts as both pro- and anti-inflammatory signaling molecules based on its concentration and the activation stage of the target cells [44]. In cardiomyocytes, $15d\text{-}PGJ_2$ showed a pro-inflammatory response that promotes the upregulation of tumor necrosis factor alpha (TNFα) [46]. During pre-termed labor, $15d\text{-}PGJ_2$ resulted in the increased production of $PGE_2$ in both vaginal and amnion epithelial cells, along with increasing levels of interleukin-8 (IL-8) in the amnion epithelial cells, which illustrated its pro-inflammatory function [45]. Moreover, $15d\text{-}PGJ_2$ production is induced by the upregulation of COX-2 in severely inflamed mammalian cells for the resolution of the inflammatory response as a negative feedback loop to terminate the inflammation [47]. Therefore, it is unclear at this point whether $15d\text{-}PGJ_2$ functions as a pro- or anti-inflammatory signaling molecule in crustaceans.

## The regulatory mechanisms of the eicosanoid biosynthesis at the transcriptional, translational, and post-translational levels

Although EHP infection altered the levels of eicosanoids and increased the levels of LvCOX and LvPGFS enzymes in shrimp hepatopancreas, little impact was observed on the transcriptional levels of eicosanoid biosynthesis genes. The lack of responses at the transcription levels could be due to multiple reasons. First, the increased substrate availability could lead to increased production of eicosanoids without affecting the eicosanoid biosynthesis gene transcripts. Second, low mRNA stability may increase the turn-over rate of the eicosanoid biosynthesis gene transcripts. This speculation is based on a study on influenza virus infection, which typically induced the eicosanoid biosynthesis pathway. However, the influenza infection resulted in a reduction in *COX* transcription levels by decreasing the stability of *COX* mRNA [48]. On the other hand, a separate study revealed that the upregulation of $PGE_2$ was required to increase the *COX* mRNA stabilization [49]. As the UHPLC-HRMS/MS analysis could not detect $PGE_2$ in the shrimp gastrointestinal tract, low mRNA stability could be the culprit for the lack of changes at the transcription levels of the eicosanoid biosynthesis genes. Third, the enzymatic activity in the eicosanoid biosynthesis pathway is also regulated post-translationally. In the soft coral *Gersemia fruticosa*, the *N*-glycosylation requirement has been demonstrated in COX-A and COX-B sequences, which contained four and six potential glycosylation sites, respectively [50]. In *P. monodon*, COX is *N*-glycosylated at three glycosylation

sites, namely N79, N170, and N424 [51]. Mutations at N170 or N424 completely disrupted the production of PGF$_{2\alpha}$, which demonstrated the requirement of *N*-glycosylation for the COX enzymatic function in shrimp. As *P. monodon* and *L. vannamei* share a high degree of sequence similarity in their genes, we believe that the COX enzyme in *L. vannamei* also requires *N*-glycosylation for the activation of its enzymatic activity. Moreover, other enzymes in the eicosanoid biosynthesis pathway, including cytosolic prostaglandin E synthase and prostaglandin D synthase, also require post-translational modifications, such as phosphorylation and glycosylation, for their enzymatic function [52,53]. Therefore, EHP infection may activate the eicosanoid biosynthesis pathway at the post-translational levels rather than at the transcriptional levels.

### The impact of gut microbiota on eicosanoid biosynthesis

Due to the presence of gut microbiota in the shrimp gastrointestinal tract, it is possible that the gut microbiome contributed to the increasing levels of eicosanoids observed in this study. Although eicosanoids cannot be synthesized by bacteria [54], the fungal population in the shrimp gastrointestinal tract may contribute to the eicosanoid production [55,56]. The secretion of eicosanoids by pathogenic fungi showed beneficial effects for the fungal survival against the host's immune response and promoted the intestinal colonization [57,58]. However, it is not possible to distinguish the eicosanoids that are produced by the host compared to the fungal population with the currently available technology. As the host is a more robust producer of eicosanoids than fungi, we believe that any changes observed in the levels of eicosanoids in the gastrointestinal tract likely stemmed from the eicosanoid biosynthesis within the shrimp cells rather than from the fungi present in these organs.

### Applications for shrimp farms

Due to the devastating impact of EHP infection on shrimp aquaculture, it is crucial to detect EHP infection early so that shrimp farmers can undertake preventative measures. However, EHP was not consistently detected during the initial stage of infection [59]. The difference in shrimp sizes due to severe growth retardation was apparent only after 2–3 months of infection [5], which resulted in the loss of resources and time for the shrimp farmers. Several factors, including the amounts of microsporidian present, water quality, and shrimp health, contributed to the inconsistency of EHP detection [60,61]. Although the current method of EHP detection shows a consistent positive result at 10 days post-infection [32], our data revealed that inflammation occurred in the shrimp gastrointestinal tract at 7 days post-infection. Therefore, we believe that the detection of PGF$_{2\alpha}$ in the shrimp intestine can be used as an additional biomarker to confirm a positive diagnosis of EHP infection. Early detection with a combination of biomarkers and qPCR could benefit future preventative efforts in farm management.

### Conclusion

EHP infection resulted in differential responses of the eicosanoid biosynthesis pathway in shrimp hepatopancreas, stomach, and intestine. On day 7 of the cohabitation experiment, PGF$_{2\alpha}$ biosynthesis was upregulated in the hepatopancreas of EHP-infected shrimp, as shown by increasing levels of ARA and increasing immunostaining signals of LvCOX and LvPGFS enzymes. The increasing levels of ARA, EPA, 12-HETE, and 12-HEPE were observed in shrimp stomach, while the levels of EPA, PGF$_{2\alpha}$, HETEs, and HEPEs also increased in shrimp intestine. As the EHP infection progressed to day 21, the levels of ARA, PGF$_{2\alpha}$, and 15d-PGJ$_2$ increased along with higher immunostaining signals of LvCOX and LvPGFS in the hepatopancreas of the EHP-infected shrimp compared to those of the control shrimp. On the other hand, HETEs and HEPEs were suppressed in the hepatopancreas of EHP-infected shrimp. EHP-infected samples collected on day 21 also contained higher levels of ARA, EPA, prostaglandins, HETEs, and HEPEs in the stomach of EHP-infected shrimp. Lastly, although the levels of eicosanoids in shrimp intestine returned to the baseline, the levels of EPA remained higher in the EHP-infected shrimp compared to the control shrimp. These data indicate that the eicosanoid biosynthesis pathway differentially responds to EHP infection based on the organ type and the duration of infection.

## Supporting information

**S1 Table.** Primer sequences and PCR conditions for the quantitative real-time PCR analysis in *L. vannamei*.
(DOCX)

**S1 Data.** qPCR analysis of SWP genes in hepatopancreas samples obtained from the cohabitation experiment.
(XLSX)

**S2 Data.** AUC ratio of eicosanoids and PUFAs to 12(S)-HETE-$d_8$ detected in the shrimp gastrointestinal tract using UHPLC-HRMS/MS.
(XLSX)

**S3 Data.** Semi-quantitative analysis of LvCOX and LvPGFS immunoreactive signals from shrimp hepatopancreases.
(XLSX)

**S4 Data.** Western blot analysis of shrimp gills using anti-COX and anti-LvPGFS antibodies to demonstrate substrate specificity.
(PDF)

**S5 Data.** qPCR analysis of *LvcPLA2, LvCOX,* and *LvPGFS* in hepatopancreases, stomachs, and intestines collected from the cohabitation experiment.
(XLSX)

## Acknowledgments

We thank Dr. Naoki Kabeya, Dr. Suparat Taengchaiyaphum, and Dr. Kallaya Sritunyalucksana for their kind suggestions. We also thank the members of the BioAssay Research Team for their support.

## Author contributions

**Conceptualization:** Wananit Wimuttisuk, Siriwan Khidprasert.

**Data curation:** Wananit Wimuttisuk, Pisut Yotbuntueng, Pacharawan Deenarn.

**Formal analysis:** Pisut Yotbuntueng, Pacharawan Deenarn, Punsa Tobwor, Kamonluk Kittiwongpukdee, Surasak Jiemsup, Rapeepun Vanichviriyakit.

**Funding acquisition:** Wananit Wimuttisuk.

**Investigation:** Wananit Wimuttisuk, Pisut Yotbuntueng, Pacharawan Deenarn, Punsa Tobwor, Kamonluk Kittiwongpukdee, Surasak Jiemsup, Rapeepun Vanichviriyakit, Chanadda Kasamechotchung, Natthinee Munkongwongsiri.

**Methodology:** Wananit Wimuttisuk, Rapeepun Vanichviriyakit, Chanadda Kasamechotchung, Suganya Yongkiettrakul, Natthinee Munkongwongsiri, Siriwan Khidprasert.

**Project administration:** Wananit Wimuttisuk.

**Resources:** Wananit Wimuttisuk.

**Validation:** Pisut Yotbuntueng, Pacharawan Deenarn, Punsa Tobwor, Surasak Jiemsup.

**Visualization:** Pisut Yotbuntueng, Pacharawan Deenarn, Punsa Tobwor, Kamonluk Kittiwongpukdee.

**Writing – original draft:** Wananit Wimuttisuk.

**Writing – review & editing:** Wananit Wimuttisuk, Pisut Yotbuntueng, Pacharawan Deenarn, Punsa Tobwor, Kamonluk Kittiwongpukdee, Surasak Jiemsup, Rapeepun Vanichviriyakit, Chanadda Kasamechotchung, Suganya Yongkiettrakul, Natthinee Munkongwongsiri, Vanicha Vichai.

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
