## [Decision Letter · Decision Letter 0]

21 Jun 2025

Dear Dr. Wimuttisuk,

Thank you for submitting your manuscript to PLOS ONE. After careful consideration, we feel that it has merit but does not fully meet PLOS ONE’s publication criteria as it currently stands. Therefore, we invite you to submit a revised version of the manuscript that addresses the points raised during the review process.

1. This manuscript not technically sound, and the data cannot support the conclusions. PLOS ONE is designed to communicate primary scientific research, and welcome submissions in any applied discipline that will contribute to the base of scientific knowledge. But this manuscript not adhere to the criteria for scientific research article that results show not sufficient to support the conclusion.

2. The revised manuscript needs to address each of the comments of the reviewers.

We look forward to receiving your revised manuscript.

Kind regards,

Tzong-Yueh Chen, Ph.D.

Academic Editor

PLOS ONE

Journal Requirements:

2. To comply with PLOS ONE submissions requirements, in your Methods section, please provide additional information regarding the experiments involving animals and ensure you have included details on methods of anesthesia and/or analgesia.

Reviewers' comments:

Reviewer's Responses to Questions

**Comments to the Author**

1. Is the manuscript technically sound, and do the data support the conclusions?

Reviewer #1: Partly

Reviewer #2: Yes

Reviewer #3: Yes

2. Has the statistical analysis been performed appropriately and rigorously?

Reviewer #1: Yes

Reviewer #2: Yes

Reviewer #3: Yes

3. Have the authors made all data underlying the findings in their manuscript fully available?

Reviewer #1: Yes

Reviewer #2: Yes

Reviewer #3: Yes

4. Is the manuscript presented in an intelligible fashion and written in standard English?

Reviewer #1: Yes

Reviewer #2: Yes

Reviewer #3: Yes

Reviewer #1: This study investigates the relationship between the eicosanoid biosynthesis pathway and Enterocytozoon hepatopenaei (EHP) infection in Litopenaeus vannamei, focusing on the analysis of EHP infection status, immunohistochemical staining, related metabolites in the pathway, and gene expression. While the study addresses an important topic, several key aspects require clarification and revision prior to publication. The following comments are provided for the authors' consideration:

1. The experimental design includes two groups: one comprising healthy, uninfected shrimp and the other consisting of EHP-infected shrimp. However, the manuscript only details the source of the healthy shrimp. It is essential to describe how the EHP-infected shrimp were obtained, were they experimentally infected in the laboratory, or sourced from commercial farms where EHP infection had been confirmed? This information is critical for understanding the infection protocol and for ensuring the reproducibility of the study.

2. A detailed description of the experimental aquaculture system is necessary. This should include tank dimensions, water circulation setup and flow rate, and how water quality was monitored and controlled. Additionally, all equipment used (e.g., filtration units, water quality monitoring systems) must be clearly identified, including brand name, model, and country of manufacture.

3. Since EHP is highly transmissible, it is imperative to describe the biosafety measures implemented during the experiment. Were standard operating procedures (SOPs) in place to prevent cross-contamination? A description of these biosafety protocols will strengthen the methodological rigor of the study.

4. The primers used for EHP detection by PCR should be explicitly listed. The manuscript should also include the standard curve for the assay and discuss the amplification efficiency of the primers. This is essential for evaluating the reliability and reproducibility of EHP detection. Furthermore, complete PCR conditions must be provided to allow replication of the results.

5. Given that EHP primarily infects and damages the hepatopancreas, the rationale for selecting the stomach and intestine for eicosanoid metabolite and gene expression analyses needs to be clearly explained. A strong justification is required to support the relevance of these tissues in the context of EHP infection and eicosanoid pathway modulation.

6. If the stomach and intestine were included due to suspected involvement in the infection or metabolic response, the manuscript should provide evidence of EHP presence in these tissues, such as PCR results or histological confirmation. Moreover, corresponding histological sections should be included to assess whether these tissues show pathological changes indicative of infection or damage, thus supporting the hypothesis that EHP may affect organs beyond the hepatopancreas.

7. Since the intestinal tract harbors a diverse microbial community capable of synthesizing lipid-derived metabolites, it is crucial to address how the study distinguished eicosanoids produced by shrimp tissues from those potentially synthesized by gut microbiota. This point is particularly important and warrants in-depth discussion or experimental clarification.

8. The manuscript must specify whether monoclonal antibodies specific to L. vannamei COX and PGFS were used. If not, the authors should provide clear evidence to validate the specificity of the antibodies, demonstrating that the observed positive reactions indeed reflect the presence of shrimp-derived COX and PGFS. Additional controls or validation data should be included to substantiate the antibody specificity.

Reviewer #2: This manuscript presents a detailed and technically sound investigation into how Litopenaeus vannamei responds to Enterocytozoon hepatopenaei (EHP) infection through changes in the eicosanoid biosynthesis pathway. The authors employ a multi-level approach—combining qPCR, UHPLC-HRMS/MS, and immunohistochemistry—to capture molecular and biochemical changes across several tissues and time points. The findings are certainly relevant to both crustacean immunology and aquaculture health management, and the identification of PGF2α and ARA as potential biomarkers is intriguing. However, there are a few important issues that limit the strength and clarity of the conclusions, and I would encourage the authors to address them more directly.

Major Comments:

1. The use of a cohabitation model for EHP infection is understandable, but I noticed that not all tanks showed qPCR-confirmed infection at the early time point (day 7)—specifically, tanks 4 and 5 had undetectable SWP levels. This inconsistency makes the grouping of "early-infected" shrimp a bit problematic. It might be helpful to either restrict downstream analyses to tanks with confirmed infection or at least stratify the data accordingly.

2. The central conclusion—that the eicosanoid changes reflect inflammation—is plausible but not convincingly supported by the current data. The study relies heavily on analogies with mammalian systems, yet no shrimp-specific inflammatory markers (e.g., immune cell infiltration, ROS, or cytokine analogs) are included. It would strengthen the paper to either incorporate such markers or soften the language regarding inflammation.

3. One interesting point is the proposed pro-inflammatory role of 15d-PGJ2 in shrimp, which contrasts with its well-characterized anti-inflammatory role in mammals. I think this idea has potential, but it deserves a more thorough discussion or supporting data to avoid appearing speculative.

4. There's a noticeable disconnect between the gene expression data and the biochemical/immunohistochemical results—most notably, key prostaglandin pathway genes showed minimal transcriptional changes despite significant alterations at the metabolite and protein levels. This discrepancy isn’t fully addressed in the discussion and could leave readers with questions about regulatory mechanisms.

5. Given the minimal changes observed at the transcript level, it would be worthwhile to expand the discussion on possible post-transcriptional regulation, protein stability, or enzyme activity shifts. These may help explain the observed metabolic changes more convincingly.

6. Lastly, many eicosanoids identified in this study can have both pro- and anti-inflammatory functions, depending on context. The interpretation here sometimes leans too heavily on mammalian literature, which might not translate directly to shrimp. I'd recommend being more cautious when assigning functional roles, and explicitly noting where assumptions are being made due to a lack of invertebrate-specific data.

Overall, this is a solid piece of research with good technical execution and potential practical applications. That said, I think the conclusions would be significantly strengthened with a more careful handling of infection confirmation, clearer linkage between omics layers, and a more nuanced discussion of eicosanoid function in crustaceans.

Reviewer #3: The authors investigated the changes in the eicosanoid biosynthesis pathway in the gastrointestinal tract of EHP-infected Litopenaeus vannamei, and they suggested that the eicosanoid biosynthesis pathway in response to early and late stages of EHP infection and implicates that inflammation is part of the host-pathogen interactions in crustaceans. Before being considered to be accepted, the present manuscript was suggested to state the potential influence among the treatments of uninfected shrimp at day 0, 7 and 21 such as shown in figure 3, 4, 5, 8 and 9.

**Do you want your identity to be public for this peer review?** For information about this choice, including consent withdrawal, please see our Privacy Policy

Reviewer #1: No

Reviewer #2: No

Reviewer #3: No

---

## [Author Response · Author response to Decision Letter 1]

19 Aug 2025

5. Review Comments to the Author

Reviewer #1: This study investigates the relationship between the eicosanoid biosynthesis pathway and Enterocytozoon hepatopenaei (EHP) infection in Litopenaeus vannamei, focusing on the analysis of EHP infection status, immunohistochemical staining, related metabolites in the pathway, and gene expression. While the study addresses an important topic, several key aspects require clarification and revision prior to publication. The following comments are provided for the authors' consideration:

Answer: We thank the reviewer for the comments for improving the quality of the manuscript. We have addressed your comments as follows:

1. The experimental design includes two groups: one comprising healthy, uninfected shrimp and the other consisting of EHP-infected shrimp. However, the manuscript only details the source of the healthy shrimp. It is essential to describe how the EHP-infected shrimp were obtained, were they experimentally infected in the laboratory, or sourced from commercial farms where EHP infection had been confirmed? This information is critical for understanding the infection protocol and for ensuring the reproducibility of the study.

Answer: The EHP-infected shrimp used to induce EHP infection in the cohabitation experiment was obtained from commercial farms where EHP infection has been confirmed. We have added this information to the materials and methods section (line 136-138).

2. A detailed description of the experimental aquaculture system is necessary. This should include tank dimensions, water circulation setup and flow rate, and how water quality was monitored and controlled. Additionally, all equipment used (e.g., filtration units, water quality monitoring systems) must be clearly identified, including brand name, model, and country of manufacture.

Answer: We have added a detailed description of the aquaculture system (i.e. tank dimensions, water circulation setup and flow rate) in the materials and methods section (line 123-131). The water quality maintenance parameters have already been provided in the original manuscript. Additional information on the brand name, model, and country for the water monitoring equipment is provided in the revised manuscript (line 154-160). As we would like to not remove the EHP spores from the rearing water, the water filtration unit was not used but rather the filtering cloth was used to remove large debris in the water circulation (line 151-154).

3. Since EHP is highly transmissible, it is imperative to describe the biosafety measures implemented during the experiment. Were standard operating procedures (SOPs) in place to prevent cross-contamination? A description of these biosafety protocols will strengthen the methodological rigor of the study.

Answer: We followed the SOP to prevent cross-contamination between the experimental groups. Detailed descriptions are added in the materials and methods section (line 145 – 150). The diagram illustrating the area in which the cohabitation experiment was conducted is shown in Fig 1.

4. The primers used for EHP detection by PCR should be explicitly listed. The manuscript should also include the standard curve for the assay and discuss the amplification efficiency of the primers. This is essential for evaluating the reliability and reproducibility of EHP detection. Furthermore, complete PCR conditions must be provided to allow replication of the results.

Answer: We have added the primer sequence, the complete PCR conditions, and the amplification efficiency in the materials and methods section (line 182-195). The amplification efficiency and the implication on EHP detection were discussed (line 547-559).

5. Given that EHP primarily infects and damages the hepatopancreas, the rationale for selecting the stomach and intestine for eicosanoid metabolite and gene expression analyses needs to be clearly explained. A strong justification is required to support the relevance of these tissues in the context of EHP infection and eicosanoid pathway modulation.

Answer: The digestive tract of shrimp runs from the mouth to the stomach, midgut, and hindgut. Additionally, the hepatopancreas directly connects to both the stomach, the anterior midgut cecum, and the midgut itself. These connections establish a real link between these organs—the stomach, hepatopancreas, and intestine. These channels allow pathogens and small molecules to travel between the organs. Although EHP infects and completes its life cycle exclusively in hepatopancreatic cells, the infected cells release spores within the hepatopancreas that can move along the digestive tract. Even small molecules released from infected cells can affect cells along the alimentary canal, which is the reason that we evaluated the alteration of the eicosanoid pathway in the stomach, hepatopancreas, and intestine to get a complete picture.

The digestive tract of shrimp anatomically extends from the mouth to the stomach, then through the midgut and hindgut, forming a continuous pathway for digestion and nutrient absorption. The hepatopancreas, a vital organ involved in digestion, is directly connected to the stomach, the anterior midgut cecum, and the midgut via specialized channels (please see the histological image below; MS submitted). These anatomical connections create a functional network that facilitates the transfer of small molecules, nutrients, and potentially pathogens between these organs, enabling coordinated physiological processes. Although EHP infects and completes its life cycle within hepatopancreatic cells, its spores can be released into the hepatopancreatic tubules and then move along the intestine. Moreover, even molecules such as metabolites or cytokines released from infected cells can influence the cellular activity in different sections of the digestive system. Therefore, to understand the full impact of EHP infection, we investigated changes in the eicosanoid pathway in the stomach, hepatopancreas, and intestine.

Figure 1 Histological image showing the anatomical connection between the stomach, hepatopancreas, and midgut (MS submitted).

Our reasoning is also supported by recent literature, in which the presence of EHP spores were detected in shrimp stomachs and intestines (Ni et al. 2025; Yuanlae et al. 2025; Zhou et al. 2020). More specifically, a study by Ni et al. (2024) revealed the presence of granulocytes in the stomach with low copy number of EHP and a substantial infiltration of hemocytes is within the connective tissue of the pyloric stomach with low and high EHP copy number, respectively.

In the low-copy pyloric stomach, there is a presence of granulocytes in the stomach with low copy number of EHP. In the high-copy pyloric stomachs, a substantial infiltration of hemocytes is observed within the connective tissue, with these cells appearing bright red or pink under staining. Granulation tissue typically appears red or purplish in Masson staining. In both high-copy pyloric and low-copy cardiac stomachs, granulation tissue, indicative of acute inflammation or tissue repair following trauma, was observed. In regions of severe EHP infection, such as the low-copy pyloric and high-copy cardiac stomachs, aggregates of microsporidia are evident.

We have added the justification for the examination of the eicosanoid biosynthesis pathway in shrimp stomach and intestine in the introduction section based on additional literature review on the inflammatory response found in shrimp stomachs and intestines during EHP infection (line 68-74) and again in the discussion section (Line 561-588). Additionally, we have provided further explanation and rationale to the Reviewer #1 as follows.

6. If the stomach and intestine were included due to suspected involvement in the infection or metabolic response, the manuscript should provide evidence of EHP presence in these tissues, such as PCR results or histological confirmation. Moreover, corresponding histological sections should be included to assess whether these tissues show pathological changes indicative of infection or damage, thus supporting the hypothesis that EHP may affect organs beyond the hepatopancreas.

Answer: There was a limitation in this study for the histological analysis to demonstrate the presence of EHP spores in shrimp stomachs and intestines. Due to the small shrimp size used in this study, we were unable to prepare the slides that sectioned the stomachs and shrimp intestines. Additionally, we have used up all of our samples during the UHPLC-HRMS/MS analysis, so we could not perform the qPCR to detect the presence of EHP spores in these organs. However, the presence of EHP in shrimp stomachs and intestines, as well as the pathological changes that are indicative of the infection and damage have been shown by Yuanlae et al. (2024) and Ni et al. (2024). We have included this information in the discussion section as previously mentioned (Line 565-588)

7. Since the intestinal tract harbors a diverse microbial community capable of synthesizing lipid-derived metabolites, it is crucial to address how the study distinguished eicosanoids produced by shrimp tissues from those potentially synthesized by gut microbiota. This point is particularly important and warrants in- depth discussion or experimental clarification.

Answer: A study by Gulbis et al (1981) established that eicosanoids cannot be synthesized by bacteria, thus ruling out the possibility that gut bacteria is responsible for the upregulation of eicosanoids in shrimp gastrointestinal tract. Nevertheless, studies in mice indicate that Candida albicans and Cryptococcus neoformans can colonize in mouse intestines and synthesize eicosanoids in the presence of ARA (Tan et al. 2019; Nover et al. 2001). As a result, it is possible that the fungal population found in shrimp gastrointestinal tract may be responsible for some of the eicosanoid production. However, based on the amounts of intestinal tissues vs. the fungal population present in shrimp intestines, most of the eicosanoids are likely produced from the host. Additionally, we have performed extensive literature review and have not come across currently available technology that can distinguish the eicosanoids produced by gut mycobiota vs. those synthesized by the host. We have included this discussion (line 672-683)

8. The manuscript must specify whether monoclonal antibodies specific to L. vannamei COX and PGFS were used. If not, the authors should provide clear evidence to validate the specificity of the antibodies, demonstrating that the observed positive reactions indeed reflect the presence of shrimp-derived COX and PGFS. Additional controls or validation data should be included to substantiate the antibody specificity.

Answer: The antibody against COX enzyme was a commercially available anti-cyclooxygenase-1 (COX-1) antibody (Cat No. ab244261, AbCAM, United Kingdom) (Line 247-249) that targeted against the C-terminal section of COX enzyme. Sequence alignment revealed that the COX epitope used to raise this polyclonal antibody were somewhat conserved toward the C-terminal sequence of the L. vannamei COX (S4 Data A). Additionally, Western blot analysis was performed to determine the presence of human COX in 293T cells as a positive control, which showed both the glycosylated (black arrow) and unglycosylated form of COX (white arrow) (S4 Data B). Similarly, the Western blot analysis of shrimp gills (S4 Data C) also revealed the presence of LvCOX band at approximately 74 kDa, which was higher than the expected size at 69.5 kDa. It was deduced that the LvCOX was also glycosylated, which resulted in a slightly higher molecular weight than the predicted protein size.

The antibody against PGFS was a polyclonal antibody that was raised against LvPGFS as stated in the materials and method (Line 249-251). A Western blot analysis of shrimp gills collected from the cohabitation experiment (D0, D0 control, and D0 EHP-infected shrimp) were tested for the presence of LvPGFS enzyme using anti-LvPGFS antibodies (S4 Data E and F). Although the predicted size of LvPGFS is 35.89 kDa, the detected LvPGFS band was slightly higher. This is also due to N-glycosylation of this enzyme, which caused slight increase in the protein size in the Western blot analysis.

Reviewer #2: This manuscript presents a detailed and technically sound investigation into how Litopenaeus vannamei responds to Enterocytozoon hepatopenaei (EHP) infection through changes in the eicosanoid biosynthesis pathway. The authors employ a multi-level approach—combining qPCR, UHPLC-HRMS/MS, and immunohistochemistry—to capture molecular and biochemical changes across several tissues and time points. The findings are certainly relevant to both crustacean immunology and aquaculture health management, and the identification of PGF2α and ARA as potential biomarkers is intriguing. However, there are a few important issues that limit the strength and clarity of the conclusions, and I would encourage the authors to address them more directly.

Major Comments:

1. The use of a cohabitation model for EHP infection is understandable, but I noticed that not all tanks showed qPCR-confirmed infection at the early time point (day 7)—specifically, tanks 4 and 5 had undetectable SWP levels. This inconsistency makes the grouping of "early-infected" shrimp a bit problematic. It might be helpful to either restrict downstream analyses to tanks with confirmed infection or at least stratify the data accordingly.

Answer: We thank the reviewer for pointing this out. Initially, we were thinking about excluding samples 4 and 5 but were afraid that this would cause a bias in the study. In the revised manuscript, we have re-calculated the analysis by excluding the data from tanks 4 and 5 in the early infection group based on the reviewer’s suggestion. The new results can be found in Figs 3, 4, 5, 8 and 9 of the revised manuscript along with new description for the results in the corresponding section.

2. The central conclusion—that the eicosanoid changes reflect inflammation—is plausible but not convincingly supported by the current data. The study relies heavily on analogies with mammalian systems, yet no shrimp-specific inflammatory markers (e.g., immune cell infiltration, ROS, or cytokine analogs) are included. It would strengthen the paper to either incorporate such markers or soften the language regarding inflammation.

Answer: We have tried examining changes in the levels of interleukins using the enzyme immunoassays against IL-1 and IL-8, however, these enzyme immunoassays did not detect any interleukin in shrimp samples, suggesting that the interleukin protein sequence in L. vannamei may be very different from those in mammals (line 583-588). We have also added results from other studies on the EHP-infected shrimp that showed immune cell infiltration in shrimp gastrointestinal tract into the introduction (line 68-74) and discussion sections (line 561-588) of this manuscript to help strengthen the paper. Moreover, we have softened the language regarding the direct tie between inflammation and the eicosanoid biosynthesis pathway and only mentioned that it is a known markers for pro- and anti-inflammatory response as to reduce over reaching for the interpretation of the data in this manuscript.

3. One interesting point is the proposed pro-inflammatory role of 15d-PGJ2 in shrimp, which contrasts with its well-characterized anti-inflammatory role in mammals. I think this idea has potential, but it deserves a more thorough discussion or supporting data to avoid appearing speculative.

Answer: We have added more discussion regarding the nuance of the 15d-PGJ2 function in the pro- and anti-inflammatory signaling molecules, which depend on the target cells and the dosages of 15d-PGJ2 present in the tissue (line 629-640). Nevertheless, there is no direct evidence to support this finding in crustaceans. As it is also not possible to culture shrimp cell lines at the moment, the experiment required to further investigate the function of 15d-PGJ2 to obtain more supporting data is currently not available.

4. There's a noticeable disconnect between the gene expression data and the biochemical/immunohistochemical resu

---

## [Decision Letter · Decision Letter 1]

9 Sep 2025

Dear Dr. Wimuttisuk,

Thank you for submitting your manuscript to PLOS ONE. After careful consideration, we feel that it has merit but does not fully meet PLOS ONE’s publication criteria as it currently stands. Therefore, we invite you to submit a revised version of the manuscript that addresses the points raised during the review process.

1. This manuscript not technically sound, and the data cannot support the conclusions. PLOS ONE is designed to communicate primary scientific research, and welcome submissions in any applied discipline that will contribute to the base of scientific knowledge. But this manuscript not adhere to the criteria for scientific research article that results show not sufficient to support the conclusion.

2. The revised manuscript needs to address each of the comments of the reviewers.

We look forward to receiving your revised manuscript.

Kind regards,

Tzong-Yueh Chen, Ph.D.

Academic Editor

PLOS ONE

Journal Requirements:

Reviewers' comments:

Reviewer's Responses to Questions

**Comments to the Author**

Reviewer #1: All comments have been addressed

Reviewer #2: All comments have been addressed

Reviewer #3: (No Response)

2. Is the manuscript technically sound, and do the data support the conclusions?

Reviewer #1: Yes

Reviewer #2: Yes

Reviewer #3: Partly

3. Has the statistical analysis been performed appropriately and rigorously?

Reviewer #1: Yes

Reviewer #2: Yes

Reviewer #3: Yes

4. Have the authors made all data underlying the findings in their manuscript fully available?

Reviewer #1: Yes

Reviewer #2: Yes

Reviewer #3: No

5. Is the manuscript presented in an intelligible fashion and written in standard English?

Reviewer #1: Yes

Reviewer #2: Yes

Reviewer #3: Yes

Reviewer #1: All comments have been addressed. Therefore, I recommend it for accepting in your esteemed journal.

Reviewer #2: The revised manuscript shows substantial improvement over the original submission. The authors have provided additional methodological details, clarified data interpretation, and strengthened the connection between EHP infection, tissue-specific responses, and eicosanoid regulation. The study is scientifically rigorous, and the revisions address most concerns raised previously. The manuscript is now close to being suitable for publication, with only minor issues requiring attention.

1. Terminology: Ensure consistent terminology for infection stages (e.g., “early infection” vs. “day 7” and “late infection” vs. “day 21”). At times both are used interchangeably within a single paragraph, which may confuse readers.

2. Grammar and style: A few sentences remain overly long and complex, particularly in the Introduction. Breaking these into shorter, more direct statements would improve readability.

Reviewer #3: The authors responded to the comments by suggesting that the potential influence of uninfected shrimp was mostly determined by other factors such as shrimp age and rearing environment. Please provide references and include them in the section of discussion.

**Do you want your identity to be public for this peer review?** For information about this choice, including consent withdrawal, please see our Privacy Policy

Reviewer #1: No

Reviewer #2: No

Reviewer #3: No

---

## [Author Response · Author response to Decision Letter 2]

15 Sep 2025

Dear Editor and Reviewers,

We are submitting a revised manuscript titled “Differential regulation of the eicosanoid biosynthesis pathway in response to Enterocytozoon hepatopenaei infection in Litopenaeus vannamei” (PONE-D-25-23707_R2) to be considered for research article publication in PLOS One. We have modified the manuscript based on the reviewers’ comments. The point-by-point responses are provided below.

---

We've checked your submission and before we can proceed, we need you to address the following issues:

1. To comply with PLOS ONE submissions requirements, in your Methods section, please provide additional information regarding the experiments involving animals and ensure you have included details on methods of anesthesia and/or analgesia.

Answer: The information regarding anesthesia was previously in the materials and methods section under the heading of EHP cohabitation experiment. However, for clarity, we have made a separate heading for “Methods of anesthesia and/or analgesia”

Reviewer's Responses to Questions

1. If the authors have adequately addressed your comments raised in a previous round of review and you feel that this manuscript is now acceptable for publication, you may indicate that here to bypass the “Comments to the Author” section, enter your conflict of interest statement in the “Confidential to Editor” section, and submit your "Accept" recommendation.

Reviewer #1: All comments have been addressed

Reviewer #2: All comments have been addressed

Reviewer #3: (No Response)

Answer: The authors thanked all the reviewers for their comments.

2. Is the manuscript technically sound, and do the data support the conclusions?

Reviewer #1: Yes

Reviewer #2: Yes

Reviewer #3: Partly

Answer: The authors understand that the difference in the levels of eicosanoids between samples collected on day 0, 7, and 21 from the control groups was not explained to the satisfaction of Reviewer #3. We added a paragraph in the discussion section (line 594-614) with references to try to convey our findings.

3. Has the statistical analysis been performed appropriately and rigorously?

Reviewer #1: Yes

Reviewer #2: Yes

Reviewer #3: Yes

Answer: We thanked the reviewers for their comments.

4. Have the authors made all data underlying the findings in their manuscript fully available?

Reviewer #1: Yes

Reviewer #2: Yes

Reviewer #3: No

Answer: The authors have provided all the raw data for the experiment in the Supplementary data section. In this revision, we have added the statistical analysis of eicosanoids in S2 Data file in the Supplementary section and the statistical analysis of qPCR is also included in S5 Data.

5. Is the manuscript presented in an intelligible fashion and written in standard English?

Reviewer #1: Yes

Reviewer #2: Yes

Reviewer #3: Yes

Answer: We thanked the reviewers for their comments

6. Review Comments to the Author

Reviewer #1: All comments have been addressed. Therefore, I recommend it for accepting in your esteemed journal.

Answer: We thanked Reviewer#1 for his/her comments.

Reviewer #2: The revised manuscript shows substantial improvement over the original submission. The authors have provided additional methodological details, clarified data interpretation, and strengthened the connection between EHP infection, tissue-specific responses, and eicosanoid regulation. The study is scientifically rigorous, and the revisions address most concerns raised previously. The manuscript is now close to being suitable for publication, with only minor issues requiring attention.

1. Terminology: Ensure consistent terminology for infection stages (e.g., “early infection” vs. “day 7” and “late infection” vs. “day 21”). At times both are used interchangeably within a single paragraph, which may confuse readers.

Answer: Thank you for your suggestion. We have adjusted the terminology to only Day 7 and Day 21 based on the reviewer’s recommendation. The change can be found throughout the manuscript.

2. Grammar and style: A few sentences remain overly long and complex, particularly in the Introduction. Breaking these into shorter, more direct statements would improve readability.

Answer: Thank you for your suggestion. We apologize for the long sentence. We have tried to cut down the sentence in the abstract, introduction, and discussion section of the manuscript based on the reviewer’s suggestions.

Reviewer #3: The authors responded to the comments by suggesting that the potential influence of uninfected shrimp was mostly determined by other factors such as shrimp age and rearing environment. Please provide references and include them in the section of discussion.

Answer: We added a paragraph in the discussion section (line 594-614) with references to try to convey our findings. Although the aging process and rearing environments can impact the levels of eicosanoids in the organisms, these are studies that were performed in mammals. Therefore, we have adjusted the reference to explain our findings based on the accumulation of nutrients, the molting period, and the stress caused by water temperature to suit the reference that we obtained in crustaceans.

We thank both the academic editor and reviewers for their invaluable comments that help improve the quality of our manuscript. We are looking forward to hearing a positive response from you.

Wananit Wimuttisuk

wananit.wim@biotec.or.th

National Science and Technology Development Agency, Thailand

---

## [Decision Letter · Decision Letter 2]

6 Oct 2025

Differential regulation of the eicosanoid biosynthesis pathway in response to Enterocytozoon hepatopenaei infection in Litopenaeus vannamei

PONE-D-25-23707R2

Dear Dr. Wimuttisuk,

We’re pleased to inform you that your manuscript has been judged scientifically suitable for publication and will be formally accepted for publication once it meets all outstanding technical requirements.

Kind regards,

Tzong-Yueh Chen, Ph.D.

Academic Editor

PLOS ONE

Additional Editor Comments (optional):

Reviewers' comments:

Reviewer's Responses to Questions

**Comments to the Author**

Reviewer #3: All comments have been addressed

2. Is the manuscript technically sound, and do the data support the conclusions?

Reviewer #3: Yes

3. Has the statistical analysis been performed appropriately and rigorously?

Reviewer #3: Yes

4. Have the authors made all data underlying the findings in their manuscript fully available?

Reviewer #3: Yes

5. Is the manuscript presented in an intelligible fashion and written in standard English?

Reviewer #3: Yes

Reviewer #3: The revised manuscript had been improved following the comments. The manuscript can be considered to be accepted.

**Do you want your identity to be public for this peer review?** For information about this choice, including consent withdrawal, please see our Privacy Policy

Reviewer #3: No

---

## [Editor Report · Acceptance letter]

PONE-D-25-23707R2

PLOS ONE

Dear Dr. Wimuttisuk,

I'm pleased to inform you that your manuscript has been deemed suitable for publication in PLOS ONE. Congratulations! Your manuscript is now being handed over to our production team.

Kind regards,

on behalf of

Prof. Tzong-Yueh Chen

Academic Editor

PLOS ONE